# VARIABLE FORWARD REGULARIZATION TO REPLACE RIDGE IN ONLINE LINEAR REGRESSION

## ABSTRACT

*Forward regularization* (-F) Azoury & Warmuth (2001) with unsupervised knowledge was proposed to replace canonical *Ridge regularization* (-R) in online linear learners, which achieves lower relative regret bounds (Della Vecchia & Basu, 2023). However, we observe that -F cannot perform as expected in practice, even possibly losing to -R for online learning tasks. We identify two main causes for this: (1) inappropriate intervention penalty; (2) potential non-i.i.d nature in online learning, both of which result in unstable posterior distribution and optima offset of the learner. To improve these, we propose *Variable Forward regularization* (-$k$F), a more general style with -F intensity modulated by a variable $k$. We further derive -$k$F algorithm to online learning tasks, which shows holistic recursive closed-form updates and superior performance compared to both -R and -F. Moreover, we theoretically establish the relative regrets of -$k$F in online learning, showing that it has a tighter upper bound than -F in adversarial settings. We also introduce an adaptive -$k$F, termed -$k$F-*Bayes*, to curb unstable penalties caused by non-i.i.d and mitigate intractable tuning of hard $k$ based on Bayesian learning for online learning. In experiments, we adapted -$k$F and -$k$F-*Bayes* into class incremental scenario, where it realized less forgetting and non-replay. Results distinctly demonstrate the efficacy of using -$k$F and -$k$F-*Bayes*.

## 1 INTRODUCTION

Unlike offline training, the online learning (OL) scenario emphasizes progressive model updates and delivers immediate decision-making on non-stationary data/task streams. Applications of OL methodology include chatbots (Chowdhury et al., 2023), injection attack detection (Toyer et al., 2024), and recommendation systems (Jiang et al., 2024) of scenes with real-time demands. During the OL process, the backbone is expected to preserve previous knowledge without past replay and reduce learning regrets (Buening et al.). In this paper, we study linear regression, one basic element of connectionist models, equipped with a novel regularization and its characteristics in online continual learning (CL) scenarios.

*Forward regularization* (-F) proposed in (Azoury & Warmuth, 2001; Vovk, 2001) was proven to have better regret bounds compared to using canonical *Ridge regularization* (-R) in linear regression with challenging adversarial bounded observations during OL Della Vecchia & Basu (2023). Specifically, -F is realized by adding an extra Frobenius-norm penalty containing upcoming unsupervised knowledge to -R, which leads the gradients of learnable weights to be more concerned with future prediction. Besides the decrease in regret, the transductive method -F has low computation complexity and regulates the learning rate in OL. As a result, -F is of practical relevance as it enhances prediction accuracy by integrating unlabeled data.

However, we observe that -F could not perform as expected in experiments, even possibly failing to -R during OL. We argue the reason is two-fold. (1) *Improper regularization.* -F defaults the penalty intensity of the unsupervised term is constant 1.0, resulting in improper penalty intervention in special tasks. (2) *Non-i.i.d disturbance.* Premise i.i.d is not held in OL, which incurs unstable regularization due to varying distribution and batch volume. This causes residuals of empirical risk minimization (ERM) and model optimum offsets, especially in learning long task streams.

To solve the problems, we first propose the *Variable Forward regularization* (-$k$F), adding a hard threshold $k$ to control the penalty strength for different tasks. The more general -$k$F is theoretically

proven to have better performance and tighter regret bounds than -R and -F by two ways if the $k$ can be given properly. The -$k$F also generates one-shot closed-form incremental updates and variable learning rate, and has less learning dissipation. Moreover, to curb unstable penalties caused by non-i.i.d and mitigate intractable tuning of hard $k$ during OL, we improve the -$k$F to -$k$F-*Bayes* style, further enabling the soft $k$ to be adaptively determined based on Bayesian learning and estimated distribution.

Our algorithm is assessed within the framework of Continual Learning (CL), which is a subset of OL. CL involves learning a sequence of $Q$ distinct tasks or episodes, denoted as $\mathcal{T} = \{\{\mathcal{T}_1, \mathcal{T}_2, \cdots, \mathcal{T}_Q\} | \mathcal{T}_q = (\mathcal{X}_q, \mathcal{Y}_q)\}$, and includes unseen classes (French, 1999). Among the various scenarios of CL, Class Incremental Learning (CIL) poses the greatest challenge. In CIL, the learner is required to integrate and retain knowledge of all previous classes, represented as $\mathcal{Y} = \bigcup_{q=1}^{Q} \mathcal{Y}_q$ without access to task identities during testing. To overcome catastrophic forgetting, EWC Kirkpatrick et al. (2017) is a pioneer regularization-based work, restricting previous weights from wrong skewing by imposing penalties. Many new methods, such as Online EWC Schwarz et al. (2018), SI Zenke et al. (2017), CRNet Li & Zeng (2023), RanPAC McDonnell et al. (2024), NICE Gurbuz et al. (2024) are born recent years. However, CIL assumes task-wise i.i.d and task boundary hinting update opportunity, hardly meeting practical needs. The harsher online task-free CIL (OTCIL) is expected to generate immediate decision-making with neither i.i.d nor boundary index in each task, where one task comes in non-stationary batches rather than all at once in CIL. Related works include GEM Lopez-Paz & Ranzato (2017), GSS Aljundi et al. (2019), and DYSON (He et al., 2024). We also applied the proposed -$k$F and -$k$F-*Bayes* methodologies to OTCIL scenarios and provided an in-depth analysis.

Contributions are summarized as follows:

1. **Propose -$k$F and derive regret bound**: To control the intervention rate of unsupervised knowledge for improved performance, -F is extended to the more general -$k$F, incorporating a $k$-factored penalty. This enhancement aims to refine the learning gradients in linear regression and reduce regrets in OL. We formulate the -$k$F algorithm with variable learning rates and recursive closed-form updates, and demonstrate that it prevents learning dissipation while achieving tighter relative regret bounds compared to -F and -R.

2. **Enhance -$k$F with Bayes**: To curb unstable penalties caused by non-i.i.d and mitigate intractable tuning of $k$ of -$k$F during OL, we further propose -$k$F-*Bayes*, enabling the soft $k$ to be adaptively determined based on Bayesian learning and distribution estimation.

3. **Practical application**: We integrated -$k$F and -$k$F-*Bayes* into randomized learners and evaluated them in (OT)CIL scenarios using both tabular and image datasets. The results demonstrated the effectiveness of our methods.

## 2 PRELIMINARY AND ADVERSARIAL REGRET BOUNDS

**Lemma 1. Bregman divergence** is used to measure relative projection distance between distributions (Azoury & Warmuth, 2001). For a real-valued differentiable convex projection $G : \boldsymbol{\theta} \in \boldsymbol{\Theta} \to \mathbb{R}$, Bregman divergence $\Delta$ is defined as:

$$\Delta_G(\tilde{\boldsymbol{\theta}}, \boldsymbol{\theta}) := G(\tilde{\boldsymbol{\theta}}) - G(\boldsymbol{\theta}) - (\tilde{\boldsymbol{\theta}} - \boldsymbol{\theta})^T \nabla_{\boldsymbol{\theta}} G(\boldsymbol{\theta}), \tag{1}$$

where the $\boldsymbol{\theta}$ is a vector, and $\nabla_{\boldsymbol{\theta}}$ denotes the gradient operator on vector $\boldsymbol{\theta}$.

**Lemma 2. Bregman divergence property.**

1. The divergence is a linear operator. $\forall \mu \geq 0, \Delta_{G_1 + \mu G_2}(\tilde{\boldsymbol{\theta}}, \boldsymbol{\theta}) = \Delta_{G_1}(\tilde{\boldsymbol{\theta}}, \boldsymbol{\theta}) + \mu \Delta_{G_2}(\tilde{\boldsymbol{\theta}}, \boldsymbol{\theta})$.
2. If $G_1(\boldsymbol{\theta}) - G_2(\boldsymbol{\theta}) = \boldsymbol{\omega}^T \boldsymbol{\theta} + \upsilon, \boldsymbol{\omega} \in \mathbb{R}^{|\boldsymbol{\theta}|}, \upsilon \in \mathbb{R}$, then $\Delta_{G_1}(\tilde{\boldsymbol{\theta}}, \boldsymbol{\theta}) = \Delta_{G_2}(\tilde{\boldsymbol{\theta}}, \boldsymbol{\theta})$.

**Setup:** following the classical setups of linear regression in OL Azoury & Warmuth (2001), a representation steam $\mathcal{X} = \{\boldsymbol{x}_t\}_{t=1}^{T} \subseteq \mathbb{R}^d$ paired with corresponding observations $\mathcal{Y} = \{y_t\}_{t=1}^{T} \subseteq \mathbb{R}$ is received by the learner $f$ one-by-one in trials $t = 1, 2, ..T$. Joint domain distribution $\mathcal{X} \times \mathcal{Y} \sim \mathcal{P}$ but drifting occurs across batches. In -R, $f_t$ is updated on visible $(\boldsymbol{x}_t, y_t)$ and evolves to $f_{t+1}$, hoping to predict $\hat{y}_{t+1}$ close to $y_{t+1}$ based on $\boldsymbol{x}_{t+1}$. The oracle learner is defined as:

$$f : y_t = \boldsymbol{x}_t^T \boldsymbol{\theta}_t^* + \varepsilon_t \quad \exists \boldsymbol{\theta}_t^* \in \boldsymbol{\Theta} \subseteq \mathbb{R}^d, 1 \leq t \leq T \tag{2}$$

where the $\varepsilon_t$ is Gaussian noise, $\boldsymbol{\theta}^*$ can be learnable weights, and $\boldsymbol{\Theta}$ is weight space.

In the optimization target, the initial $G$, incurred loss on $\boldsymbol{x}_t$, and forward predictive loss are respectively designated to $U_0(\boldsymbol{\theta}) = \frac{1}{2}\boldsymbol{\theta}^T\boldsymbol{\eta}_0^{-1}\boldsymbol{\theta}$, $\mathcal{L}_t(\boldsymbol{\theta}) = \frac{1}{2}||\boldsymbol{x}_t^T\boldsymbol{\theta} - y_t||_2^2$, and $\hat{\mathcal{L}}_{t+1}(\boldsymbol{\theta}) = \frac{1}{2}||\boldsymbol{x}_{t+1}^T(\boldsymbol{\theta} - \boldsymbol{\theta}_0)||_2^2$, where $\boldsymbol{\eta}_0^{-1}$ is a symmetric positive definite matrix and $\boldsymbol{\theta}_0$ is the initial parameter.

**Lemma 3. Offline learning** refers to the learning behavior of an expert on global tasks. Online learner is contrasted to the expert by relative regret bounds. Assume solutions always exist in $\boldsymbol{\theta} \in \boldsymbol{\Theta}$:

$$\boldsymbol{\theta}_{Q+1} = argmin_{\boldsymbol{\theta}}U_{Q+1}(\boldsymbol{\theta}), \tag{3}$$

where $U_{Q+1}(\boldsymbol{\theta}) = \Delta_{U_0}(\boldsymbol{\theta}, \boldsymbol{\theta}_0) + \mathcal{L}_{1..Q}(\boldsymbol{\theta})$, $\boldsymbol{\theta}_{Q+1}$ represents the finally updated parameter for future predictions after the last $Q$-th task knowledge acquisition completed.

**Lemma 4. Online-to-offline regret bounds** are defined as the upper bounds of cumulative regrets of an online learner over those of the offline expert, which quantify the gap to the best expert and be regarded as the cost of hiding future data from the learner. The upper bound of the learner using -R:

$$\sum_{t=1}^{T}\mathcal{L}_t(\boldsymbol{\theta}_t^r) - \min_{\boldsymbol{\theta}}(\frac{1}{2}\lambda||\boldsymbol{\theta} - \boldsymbol{\theta}_0||_2^2 + \sum_{t=1}^{T}\mathcal{L}_t(\boldsymbol{\theta})) \leq 2Y_m^2 dIn(\frac{TX_m^2}{\lambda} + 1) \tag{4}$$

where $X_m = \max_{1 \leq t \leq T}\{||\boldsymbol{x}_t||_\infty\}$, $Y_m = \max_{1 \leq t \leq T}\{|y_t|, |\boldsymbol{x}_t^T\boldsymbol{\theta}_t|\}$, and $\lambda$ is the penalty factor. This is based on adversarial setups and highlights the prediction $\boldsymbol{x}_t^T\boldsymbol{\theta}_t \in [-Y_m, Y_m]$ for history value restraints.

The upper bound of the learner using -F:

$$\sum_{t=1}^{T}\mathcal{L}_t(\boldsymbol{\theta}_t^f) - \min_{\boldsymbol{\theta}}(\frac{1}{2}\lambda||\boldsymbol{\theta}||_2^2 + \sum_{t=1}^{T}\mathcal{L}_t(\boldsymbol{\theta})) \leq \frac{1}{2}Y_m^2 dIn(\frac{TX_m^2}{\lambda} + 1) \tag{5}$$

where $X_m = \max_{1 \leq t \leq T}\{||\boldsymbol{x}_t||_\infty\}$, $Y_m = \max_{1 \leq t \leq T}\{|y_t|\}$. Prediction range assumption is removed and only retains $y_t \in [-Y_m, Y_m]$. Note the -F bound is at least 4 times better than -R's.

# 3 PROPOSED METHODOLOGY AND REGRET BOUND DERIVATION

## 3.1 ONLINE LEARNING FRAMEWORK FOR $-k$F STYLE

**Theorem 1. Incremental offline learning using $-k$F.** We define the $-k$F optimization target and express an incremental offline pattern here. For $0 \leq t \leq T$, the following equations are optimized:

$$\boldsymbol{\theta}_{t+1} = argmin_{\boldsymbol{\theta}}U_{t+1}(\boldsymbol{\theta}) \tag{6}$$
$$U_{t+1}(\boldsymbol{\theta}) = \Delta_{U_0}(\boldsymbol{\theta}, \boldsymbol{\theta}_0) + \mathcal{L}_{1..t}(\boldsymbol{\theta}) + k \cdot \hat{\mathcal{L}}_{t+1}(\boldsymbol{\theta})$$

where $\hat{\mathcal{L}}$ is the estimated loss on upcoming data (reserved prior knowledge is also possible). $-k$F integrates unsupervised knowledge and affects optimization gradients (also a posterior optimum) when updating the present model. Note $\boldsymbol{\theta}_1 = argmin_{\boldsymbol{\theta}} U_1(\boldsymbol{\theta}) = \boldsymbol{\theta}_0$ when $t = 0$.

**Theorem 2. OL using $-k$F.** For $0 \leq t \leq T$, with previous $\mathcal{L}_{1..t-1}$ concentrated into Bregman divergence $\Delta U_t$, we define the OL process of $-k$F as optimizing a recursive equation without replay.

$$\boldsymbol{\theta}_{t+1} = argmin_{\boldsymbol{\theta}}\Delta_{U_t}(\boldsymbol{\theta}, \boldsymbol{\theta}_t) + \mathcal{L}_t(\boldsymbol{\theta}) + k \cdot \hat{\mathcal{L}}_{t+1}(\boldsymbol{\theta}) - k \cdot \hat{\mathcal{L}}_t(\boldsymbol{\theta}) \tag{7}$$

When $0 < k \leq 1$, it can be regarded as partial forward knowledge intervention because ERM on $\boldsymbol{x}_t$ is larger than the forward penalty of $\boldsymbol{x}_{t+1}$. Having $k > 1$ leads to an excess in forward regularization. ***Proof***: Please see Appendix A.1

## 3.2 ONLINE LEARNING ALGORITHM FOR $-k$F STYLE

**Theorem 3. OL algorithm using $-k$F.** The step-wise updates of learnable weight $\boldsymbol{\theta}$ in $-k$F follow:

$$\boldsymbol{\theta}_{t+1} = \boldsymbol{\theta}_t - \boldsymbol{\eta}_{t+1}[(\boldsymbol{x}_t\boldsymbol{x}_t^T + k \cdot \boldsymbol{x}_{t+1}\boldsymbol{x}_{t+1}^T - k \cdot \boldsymbol{x}_t\boldsymbol{x}_t^T)\boldsymbol{\theta}_t - \boldsymbol{x}_t y_t] \tag{8}$$

where $k$ is a constant, $\boldsymbol{\theta}_0 = 0$, symmetric positive defined $\boldsymbol{\eta}_0 = (\lambda \cdot \boldsymbol{I})^{-1}$, and variable learning rates $\boldsymbol{\eta}_{t+1} = (\boldsymbol{\eta}_0^{-1} + \sum_{i=1}^{t} \boldsymbol{x}_i \boldsymbol{x}_i^T + k \cdot \boldsymbol{x}_{t+1} \boldsymbol{x}_{t+1}^T)^{-1}$. The step-wise updates of $\boldsymbol{\eta}_{t+1}$ follow:

$$\boldsymbol{\eta}_{t+1}^\dagger = \boldsymbol{\eta}_t^\dagger - \boldsymbol{\eta}_t^\dagger \boldsymbol{x}_t (\boldsymbol{I} + \boldsymbol{x}_t^T \boldsymbol{\eta}_t^\dagger \boldsymbol{x}_t)^{-1} \boldsymbol{x}_t^T \boldsymbol{\eta}_t^\dagger \tag{9}$$

$$\boldsymbol{\eta}_{t+1} = \boldsymbol{\eta}_{t+1}^\dagger - \boldsymbol{\eta}_{t+1}^\dagger k \cdot \boldsymbol{x}_{t+1} (\boldsymbol{I} + k \cdot \boldsymbol{x}_{t+1}^T \boldsymbol{\eta}_{t+1}^\dagger \boldsymbol{x}_{t+1})^{-1} \boldsymbol{x}_{t+1}^T \boldsymbol{\eta}_{t+1}^\dagger$$

where $\boldsymbol{\eta}_{t+1}^\dagger = (\boldsymbol{\eta}_0^{-1} + \sum_{i=1}^{t} \boldsymbol{x}_i \boldsymbol{x}_i^T)^{-1}$ is maintained as an intermediate variable for computation. This OL process of using $-k$F is non-replay, and the Bregman divergence term changes over time.

***Proof***: Please see Appendix A.2

**Remark 1.** Our $-k$F style is more general. If $k = 0$, Theorem 2 degenerates to -R style:

$$\boldsymbol{\theta}_{t+1} = argmin_{\boldsymbol{\theta}} \Delta_{U_t}(\boldsymbol{\theta}, \boldsymbol{\theta}_t) + \mathcal{L}_t(\boldsymbol{\theta}) \tag{10}$$

$$\boldsymbol{\theta}_{t+1} = \boldsymbol{\theta}_t - \boldsymbol{\eta}_{t+1}[\boldsymbol{x}_t \boldsymbol{x}_t^T \boldsymbol{\theta}_t - \boldsymbol{x}_t y_t]$$

If $k = 1$, Theorem 2 degenerates to -F style:

$$\boldsymbol{\theta}_{t+1} = argmin_{\boldsymbol{\theta}} \Delta_{U_t}(\boldsymbol{\theta}, \boldsymbol{\theta}_t) + \mathcal{L}_t(\boldsymbol{\theta}) + \hat{\mathcal{L}}_{t+1}(\boldsymbol{\theta}) - \hat{\mathcal{L}}_t(\boldsymbol{\theta}) \tag{11}$$

$$\boldsymbol{\theta}_{t+1} = \boldsymbol{\theta}_t - \boldsymbol{\eta}_{t+1}[\boldsymbol{x}_{t+1} \boldsymbol{x}_{t+1}^T \boldsymbol{\theta}_t - \boldsymbol{x}_t y_t]$$

This further proves $-k$F style is more general and the performance of using $-k$F must not be inferior to that of -R or -F if $k$ is properly designed, and such $k$ values exist with a high probability.

**Remark 2.** We further derive several equalities before the regret bound derivation. Given $\boldsymbol{\eta}_{t+1}^{-1} = \boldsymbol{\eta}_t^{-1} + (1 - k) \cdot \boldsymbol{x}_t \boldsymbol{x}_t^T + k \cdot \boldsymbol{x}_{t+1} \boldsymbol{x}_{t+1}^T$ and $\boldsymbol{\theta}_0 = 0$, for $1 \le t \le T$ we have:

$$\boldsymbol{\theta}_{t+1} = \boldsymbol{\eta}_{t+1}(\boldsymbol{\eta}_t^{-1} \boldsymbol{\theta}_t + \boldsymbol{x}_t y_t) \tag{12}$$

$$\boldsymbol{\theta}_{t+1} = \boldsymbol{\theta}_t - \boldsymbol{\eta}_t[(\boldsymbol{x}_t \boldsymbol{x}_t^T + k \cdot \boldsymbol{x}_{t+1} \boldsymbol{x}_{t+1}^T - k \cdot \boldsymbol{x}_t \boldsymbol{x}_t^T)\boldsymbol{\theta}_{t+1} - \boldsymbol{x}_t y_t] \tag{13}$$

$$\boldsymbol{\theta}^{off} = (\boldsymbol{\eta}_0^{-1} + \sum_{i=1}^{T} \boldsymbol{x}_i \boldsymbol{x}_i^T)^{-1} \boldsymbol{\eta}_{T+1}^{-1} \boldsymbol{\theta}_{T+1} \tag{14}$$

***Proof***: Please see Appendix A.3

### 3.3 RELATIVE REGRET BOUND FOR $-k$F STYLE

**Theorem 4. Relative Regrets for $-k$F.** For any $\mathcal{X} = \{\boldsymbol{x}_t\}_{t=1}^T$ and any $\boldsymbol{\theta} \in \boldsymbol{\Theta}$, the cumulative regrets of the online learner using $-k$F style relative to the offline expert can be calculated as follows:

$$\sum_{t=1}^{T} \mathcal{L}_t(\boldsymbol{\theta}_t) - \min_{\boldsymbol{\theta} \in \boldsymbol{\theta}^{off}} (\Delta_{U_0}(\boldsymbol{\theta}, \boldsymbol{\theta}_0) + \sum_{t=1}^{T} \mathcal{L}_t(\boldsymbol{\theta})) \tag{15}$$

$$= \sum_{t=1}^{T} [\Delta_{U_{t+1}}(\boldsymbol{\theta}_t, \boldsymbol{\theta}_{t+1}) - k \cdot \hat{\mathcal{L}}_{t+1}(\boldsymbol{\theta}_t) + k \cdot \hat{\mathcal{L}}_t(\boldsymbol{\theta}_t)] + k \cdot \hat{\mathcal{L}}_{T+1}(\boldsymbol{\theta}) - \Delta_{U_{T+1}}(\boldsymbol{\theta}, \boldsymbol{\theta}_{T+1})$$

***Proof***: Please see Appendix A.4

**Theorem 5. Relative Regret Bounds for $-k$F.** The upper bound of the learner using $-k$F:

$$\mathbb{E}[\sum_{t=1}^{T} \mathcal{L}_t(\boldsymbol{\theta}_t) - \min_{\boldsymbol{\theta} \in \boldsymbol{\theta}^{off}} (\Delta_{U_0}(\boldsymbol{\theta}, \boldsymbol{\theta}_0) + \sum_{t=1}^{T} \mathcal{L}_t(\boldsymbol{\theta}))] = \frac{k}{2} Y_m^2 d In(1 + \frac{T X_m^2}{\lambda + (k-1) \cdot X_m^2}) \tag{16}$$

where $X_m = \max_{1 \le t \le T} \{||\boldsymbol{x}_t||_\infty\}$, $Y_m = \max_{1 \le t \le T} \{|y_t|\}$, and $k > 0$. Like (5), (16) still maintains the advantages of border (i.e. $\boldsymbol{x}_t^T \boldsymbol{\theta}_t \in [-Y_m, Y_m]$) removal. (16) corresponds to (5)'s form if $k = 1$.

***Proof***: Please see Appendix A.5

**Remark 3.** Let $\mathbb{E}[\sum_{t=1}^{T} \mathcal{L}_t(\boldsymbol{\theta}_t) - \min_{\boldsymbol{\theta} \in \boldsymbol{\theta}^{off}} (\Delta_{U_0}(\boldsymbol{\theta}, \boldsymbol{\theta}_0) + \sum_{t=1}^{T} \mathcal{L}_t(\boldsymbol{\theta}))] = \mathbb{E}[\Gamma]$. Assume the entire $\mathcal{X}$ respects independent identical Gaussian distribution in proof, which allows us to involve approximation $\mathbb{E}[\boldsymbol{x}_{t+1}^T \boldsymbol{\eta}_t \boldsymbol{x}_t] = 0$. Such that the $\Gamma$ is a complex random variable associated with data distribution $\mathcal{P}_\mathcal{X}$. Consequently, the regret bounds are fluctuating. How to choose the $k$ is a game problem, as shown in (16). One can explore the $\Gamma$ distribution and lead to the proper $k$'s range based on a specific data environment.

**Remark 4.** The actual $\mathbb{E}[\Gamma]$ can be lower than that in (16), as we omit some negative penalty terms in the calculation of (55). It will be complicated to obtain the proper $k$'s range by studying $\frac{1}{2}Y_m^2 dIn(\frac{TX_m^2}{\lambda}+1) - \frac{k}{2}Y_m^2 dIn(1+\frac{TX_m^2}{\lambda+(k-1)\cdot X_m^2}) \geq 0$. To avoid this, we study the growth rates of cumulative relative regrets:

$$\frac{\partial \frac{1}{2}\hat{Y}_m^2 dIn(\frac{t\hat{X}_m^2}{\lambda}+1)}{\partial t} = \frac{1}{2}\hat{Y}_m^2 d \cdot \frac{\hat{X}_m^2}{\lambda+t\hat{X}_m^2} \tag{17}$$

$$\frac{\partial \frac{k}{2}\hat{Y}_m^2 dIn(1+\frac{t\hat{X}_m^2}{\lambda+(k-1)\cdot \hat{X}_m^2})}{\partial t} = \frac{k}{2}\hat{Y}_m^2 d \cdot \frac{\hat{X}_m^2}{\lambda+(k-1)\cdot \hat{X}_m^2 + t\hat{X}_m^2} \tag{18}$$

Let (17)-(18)$> 0$, for $1 \leq t \leq T$, we have:

$$0 < k < 1 \tag{19}$$

where $\hat{X}_m = \max_{1 \leq i \leq t+1}\{||\boldsymbol{x}_i||_\infty\}$, $\hat{Y}_m = \max_{1 \leq i \leq t}\{|y_i|\}$.

In conclusion, the $k$ that makes $-k$F have a tighter regret bound than -F's exists.

## 3.4 ADAPTIVE $-k$F-*Bayes* STYLE

**Potential non-i.i.d nature.** In Lemma 3 and Theorem 1, impacts due to potential non-i.i.d are negligible as all training data is available, and the severity of penalties entirely maintains balance at time $t$. Inversely, the effects of non-i.i.d should be taken into account in Theorem 2 and 3, because we emphasize OL and CL contexts including non-i.i.d data batch, off-diagonal covariance matrix, invisible future data, and forbidden past replay. Merely setting sensitive $k$ as fixed results in penalty imbalance and offsets the learner's optima during OL/CL process.

$-k$**F uses hard threshold** $k$. The $k$ mediating contributed degrees of $k \cdot \hat{\mathcal{L}}_{t+1}(\boldsymbol{\theta}) - k \cdot \hat{\mathcal{L}}_t(\boldsymbol{\theta})$ determines performance of the learner using $-k$F style. Using gradient descent or evolutionary algorithm to adjust $k$ at each step is unreasonable due to time-consuming and non-replay, and a compromise hard $k$ instead of synchronized one can be thrown even if global optimization for $k$ is allowed.

Although we present a range of $k$ in (19), it still has a relatively loose upper limitation with respect to $k$ because we use $\hat{X}_m$ in adversarial environment and approximation in Theorem 5, which cannot generate specific values or accurate range. To improve this, we regard the Theorem 2 and Theorem 3 as Bayesian learning processes, and study how to suppress non-i.i.d impact on posterior distribution by adaptively determined $k$, based on distribution estimation and penalty balance of prior terms.

**Theorem 6. Adaptive** $-k$**F-*Bayes* style** The step-wise updates of learnable weight $\boldsymbol{\theta}$ in $-k$F-*Bayes* follow:

$$\boldsymbol{\theta}_{t+1} = \boldsymbol{\theta}_t - \boldsymbol{\eta}_{t+1}[(\boldsymbol{x}_t\boldsymbol{x}_t^T + k_{t+1}\cdot\boldsymbol{x}_{t+1}\boldsymbol{x}_{t+1}^T - k_t\cdot\boldsymbol{x}_t\boldsymbol{x}_t^T)\boldsymbol{\theta}_t - \boldsymbol{x}_t y_t] \tag{20}$$

where $k$ is a variable, $\boldsymbol{\theta}_0 = 0$, symmetric positive defined $\boldsymbol{\eta}_0 = (\lambda\cdot\boldsymbol{I})^{-1}$, and variable learning rates $\boldsymbol{\eta}_{t+1} = (\boldsymbol{\eta}_0^{-1} + \sum_{i=1}^t \boldsymbol{x}_i\boldsymbol{x}_i^T + k_{t+1}\cdot\boldsymbol{x}_{t+1}\boldsymbol{x}_{t+1}^T)^{-1}$. The step-wise updates of $\boldsymbol{\eta}_{t+1}$ follow:

$$\boldsymbol{\eta}_{t+1}^\dagger = \boldsymbol{\eta}_t^\dagger - \boldsymbol{\eta}_t^\dagger\boldsymbol{x}_t(\boldsymbol{I} + \boldsymbol{x}_t^T\boldsymbol{\eta}_t^\dagger\boldsymbol{x}_t)^{-1}\boldsymbol{x}_t^T\boldsymbol{\eta}_t^\dagger$$
$$k_{t+1} = k_t = \boldsymbol{x}_t^T\boldsymbol{\eta}_t\boldsymbol{x}_t \tag{21}$$
$$\boldsymbol{\eta}_{t+1} = \boldsymbol{\eta}_{t+1}^\dagger - \boldsymbol{\eta}_{t+1}^\dagger k_{t+1}\cdot\boldsymbol{x}_{t+1}(\boldsymbol{I} + k_{t+1}\cdot\boldsymbol{x}_{t+1}^T\boldsymbol{\eta}_{t+1}^\dagger\boldsymbol{x}_{t+1})^{-1}\boldsymbol{x}_{t+1}^T\boldsymbol{\eta}_{t+1}^\dagger$$

where $\boldsymbol{\eta}_{t+1}^\dagger = (\boldsymbol{\eta}_0^{-1} + \sum_{i=1}^t \boldsymbol{x}_i\boldsymbol{x}_i^T)^{-1}$ is maintained as an intermediate variable for computation. This OL process of using $-k$F-*Bayes* is still non-replay, can resist catastrophic forgetting, and adaptively determine the sensitive $k$ factor.

***Proof***: Please see Appendix A.6

**Remark 5.** Given $\boldsymbol{B} = \{\boldsymbol{x}_i\}_{i=1}^b$ containing $b$ samples, (66) is enhanced to:

$$k_{t+1} = \kappa \cdot (\frac{trace[(\boldsymbol{B}_{t+1}\boldsymbol{\eta}_t\boldsymbol{B}_{t+1}^T + \sigma\boldsymbol{I})^{-1}]}{b})^{-1} \tag{22}$$

$$k_t = \kappa \cdot (\frac{trace[(\boldsymbol{B}_t\boldsymbol{\eta}_t\boldsymbol{B}_t^T + \sigma\boldsymbol{I})^{-1}]}{b})^{-1}$$

# 4 EXPERIMENT

We first examined online learners with different regularization styles, namely -R, -F, -$k$F, and -$k$F-*Bayes* in numerical simulations. Then we integrated these online learners into Randomized Neural Networks (Randomized NN) (Li & Zeng, 2023; Li et al., 2024) and conducted experiments in CIL/OTCIL scenarios, including tabular and image datasets. Based on the results, we show the efficacy of -$k$F and -$k$F-*Bayes*, and the great potential of Randomized learners equipped with them.

## 4.1 CASE 1: NUMERICAL SIMULATION

The numerical scenario was generated by the setups: $T$=1000, $d = 12$, $\varepsilon_t \sim \mathcal{N}(1, 2)$, $\boldsymbol{x}_t \sim \mathcal{N}_d(\boldsymbol{6I}, \boldsymbol{\Sigma})$, $\lambda \in \{0.2, 0.5, 0.8, 1.0, 1.5, 2.0, 3.0\}$, $k \in \{0.2, 0.4, 0.6, 0.8\}$, and the regularization styles included: -R, -F, -$k$F, -$k$F-*Bayes*. We repeated the learning processes of the online learners 20 times using different random seeds for each $\lambda$. Results would be used to compute the mean and standard deviation of online regrets relative to Oracle learner (defined in (2)) in multiple trials. We assumed the learners had no prior knowledge before data came and started from $\boldsymbol{\theta} = \boldsymbol{0}$.

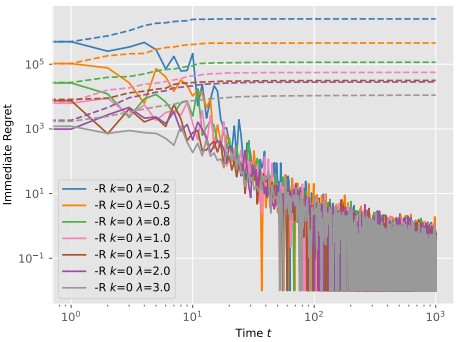

(a) Relative regrets of online learner using -R.

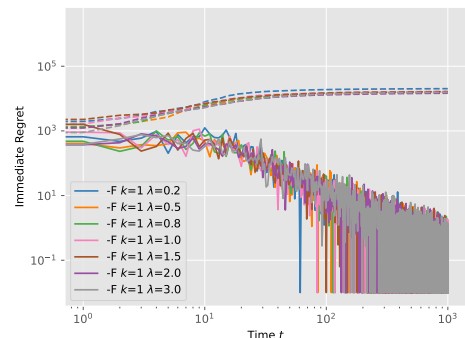

(b) Relative regrets of online learner using -F.

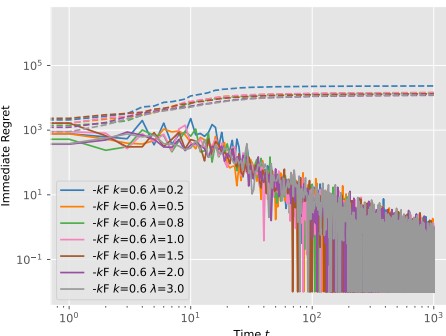

(c) Relative regrets of online learner using -0.6F.

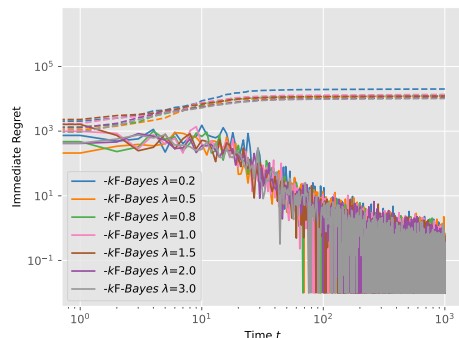

(d) Relative regrets of online learner using -$k$F-*Bayes*.

Figure 1: Relative immediate regrets (shown in solid lines) and cumulative regrets (shown in dashed lines) of online linear regression using different regularization styles. The shadows indicate ranges within one standard deviation. -0.6F represents the best performance in all tested -$k$F, one can refer to Appendix A.7 for other results.

As shown in Figure 1, learners using -R, also regarded as -0F style, suffer from low stability and high immediate regrets at all $\lambda$s. Although it converges quickly at a later stage, it causes high cumulative regrets in OL process. -F (i.e. $-1.0$F) style is more stable to -R and the cumulative error decreases significantly. Here we roughly selected several points for studying $k$'s effects. The best -0.6F style slightly reduces cumulative regrets and responds faster than -F. The effect is not obvious because such a $k$ exists but is difficult to determine and the same $k$ is poorly adapted to different

environments, i.e., it does not perform as expected for all $\lambda$s. Appendix A.7 also shows that $k$ is sensitive and an improper $k$ results in a large loss. How to set a proper $k$ is important because the entire dataset is unavailable and real-time searching is unreasonable in practice. The -$k$F-*Bayes* achieves rapid decreases of immediate regrets and lower cumulative loss with the $k$s adaptively set in OL. The advantages of -$k$F-*Bayes* will become more apparent in difficult tasks. The $k$ variation curves are shown in Figure 2

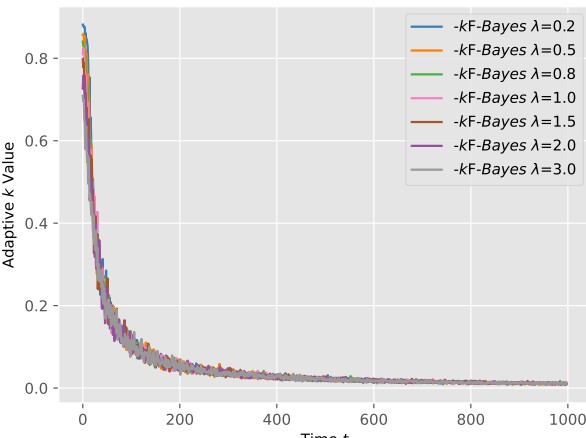

Figure 2: $k$ variation curves of online learners with -$k$F-*Bayes* in numerical simulations.

## 4.2   CASE 2: TABULAR DATASET

In order to examine linear regression learners using various regularization styles in more complex tasks, we used -$k$F and -$k$F-*Bayes* algorithms to reconstruct a state-of-the-art Randomized NN, the ensemble deep random vector functional link network (edRVFL) (Shi et al., 2021), and adapted edRVFL-$\mathcal{A}$ to CIL/OTCIL scenarios, where $\mathcal{A}$ denotes varied styles. The datasets are listed in Table 1 and each one was treated as a single class incremental problem, namely learning on [Dataset]-$|\mathcal{Y}|/|\mathcal{Y}|$ task stream.

Main methods involved in this experiment were as follows. (1) SMAC3 (Lindauer et al., 2022): we used it to configure well-behaved hyper-parameters (HP) within the same searching space for algorithms. (2) edRVFL (Shi et al., 2021): the original offline edRVFL network. (3) edRVFL-R, edRVFL-$k$F and edRVFL-$k$F-*Bayes*: our built methods for comparisons. We assumed they had no prior knowledge or update on any task, and always did one-pass training. We use $N$ and $L$ to denote edRVFL's numbers of layers and nodes per layer respectively. (4) EWC (Kirkpatrick et al., 2017): the backbone of EWC for CIL was a BP-based MLP. (5) CRNet-I (Li & Zeng, 2023): a novel CIL network based on Randomized NN and EWC. (6) DYSON (He et al., 2024): an OTCIL method using compute-and-align paradigm. Some assess metrics containing $ACC$, $BWT$, and $FWT$ are list in Appendix A.8 **Evaluation metrics**.

During SMAC3 operation, every dataset was randomly partitioned into 4-folds for consistent results, containing $60\%$ for training, $15\%$ for validation, and $25\%$ for testing in each fold. We set trial times to 200. In one trial, the algorithm configured by SMAC3 would be tested on 4-folds separately using all different random seeds on task order and network initialization. The average $acc.(T)$ on 4-folds was set to be the cost of incumbents which guided SMAC3 to optimize. Note the $k$ of -$k$F was optimized while was adaptive in -$k$F-*Bayes*.

Testset performance of the above algorithms with HP and structures optimized by SMAC3 is shown in Table 2 in Appendix A.8. Metrics are average $acc.(T)\%$ (for OTCIL), average $ACC(Q)\%$ (for CIL), and $std.\%$ of 4-fold trials. The results show that: (1) edRVFL-$k$F and edRVFL-$k$F-*Bayes* outperform other methods on most datasets; (2) although edRVFL-$k$F has favorable performance, it relies on fatal HP $k$ value optimized by SMAC3. Searching methods (e.g. PSO, grid searching) are

usually time-consuming and difficult to optimize $-k$F synchronously in CIL, while the $-k$F-*Bayes* is ready to be deployed and achieves impressive results (even a little worse than $-k$F occasionally); (3) the $-k$F style using forward unsupervised information is not worse than -R and when $k = 0$ they have similar results; (4) BP-based EWC and DYSON suffer from larger loss, and CRNet also shows undesirable capability, especially on longer task stream (e.g. letters, plant margin). (5) our methods are noticeably more stable as indicated by the low stds, and are better suited to this situation because no learning dissipation in theory. These conclusions also encourage the extension of our methods to the representative learning on image features transformed by pre-trained models (PTM).

We offer $ACC$ curves on letters-26/26 as shown in Fig. 3. The optimized HP and structures were still employed. Our methods were set to no prior knowledge and start by $\boldsymbol{\theta}_0 = 0$ and diagonal $\boldsymbol{\eta}_0$ during OTCIL, which resulted in accuracy lags at $q \leq 5$. However, the hysteresis of edRVFL-$k$F-*Bayes* is slightest compared to using -R or $-k$F, and all proposed methods arrive at the expected accuracy as more tasks are learned. EWC and CRNets yield increasing loss on long task stream, and DYSON gets around 20% more accuracy because of PTM and replay system. Note the optimization process like using SMAC3 is almost necessary for $-k$F desired performance as its $k$ requests careful setting, while is needless for $-k$F-*Bayes*.

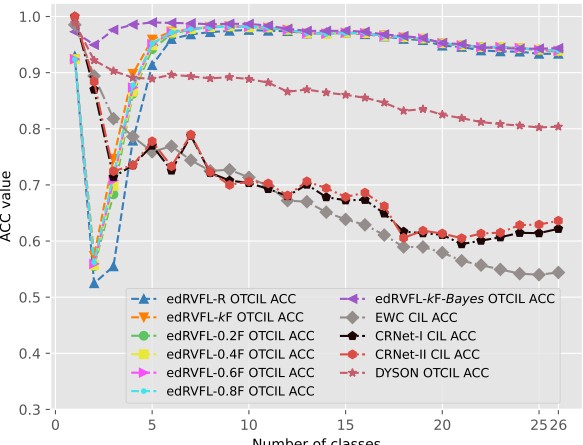

Figure 3: $ACC$ curves of methods in Table 2 (in Appendix A.8) on letters-26/26 in CIL. The edRVFL-R, edRVFL-$k$F, and edRVFL-$k$F-*Bayes* learned one class in two batches without boundary.

The incurred loss during OTCIL is shown in Fig. 4 (a), which demonstrates the superiority of edRVFL-$k$F-*Bayes* in terms of accuracy, dynamic response, and regrets. The $k$s in $-k$F were unreasonably identical in layers and time-consuming to tune, which was avoided and ready-to-use in $-k$F-*Bayes*. The dynamic adaptive process of $k$ in $-k$F-*Bayes* is shown in Fig. 4 (b), and it pays more attention to a future task compared to past ones. The $k$ of $-k$F is sensitive and affected by multiple facts, and its being in the range of the maximum and minimum $k_t$ of $-k$F-*Bayes* is interesting, which highlights SMAC3's choice and its effectiveness.

### 4.3 CASE 3: CIFAR IMAGE DATASET

For the CIFAR-100 dataset, we split it into CIFAR-100/10 to implement experiments. We studied selected methods, such as EWC (Kirkpatrick et al., 2017), CRNet (Li & Zeng, 2023) for the regularization-based group, RanPAC (McDonnell et al., 2024), NICE (Gurbuz et al., 2024), DYSON He et al. (2024) for the model-based methods, and GEM (Lopez-Paz & Ranzato, 2017), GSS (Aljundi et al., 2019) for the replay-based branch. A standard resnet-56 was employed as PTM and used to extract features from CIFAR-100, which inherited the setup in (Li & Zeng, 2023) for fair comparison, and to allow benchmarks to learn image representations. We offered the same PTM for all algorithms except when they are already equipped with one, such as for NICE and DYSON. After resnet-56, the PTM $\mathcal{F}(\cdot)$ was followed in our proposed methods for enhancement. The task

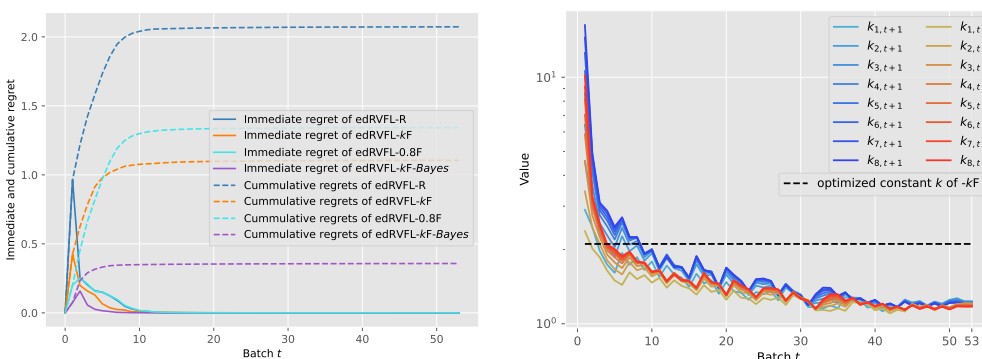

(a) The letter testset immediate and cumulative $regret(t)$ curves of edRVFL using -R, -$k$F, and -$k$F-*Bayes* in OTCIL. -$k$F-*Bayes* has the minimum loss. X-axis denotes batch, and Y-axis is regret value.

(b) $k_{l,t+1}$ and $k_{l,t}$ variation curves of -$k$F-*Bayes* in OT-CIL. X-axis denotes batch, Y-axis is the logarithmic value. Trends illustrate the effect of maintaining the balance of penalties in changing optimization targets.

Figure 4: The regrets on testset in OTCIL and adaptive $k$ variation curves of edRVFL-$k$F-*Bayes*.

order was randomly sorted and baselines were tested 10 times. No task boundary was given to OTCIL methods. Our methods still learned every task in one-pass, had no revisits, and we abode by severe no prior update before the data arrival.

The task order and choice of classes in CIFAR-100/10 were randomized. From Figure 5 and Figure 6, the proposed edRVFL-$k$F-*Bayes* still maintains an obvious advantage in performance.

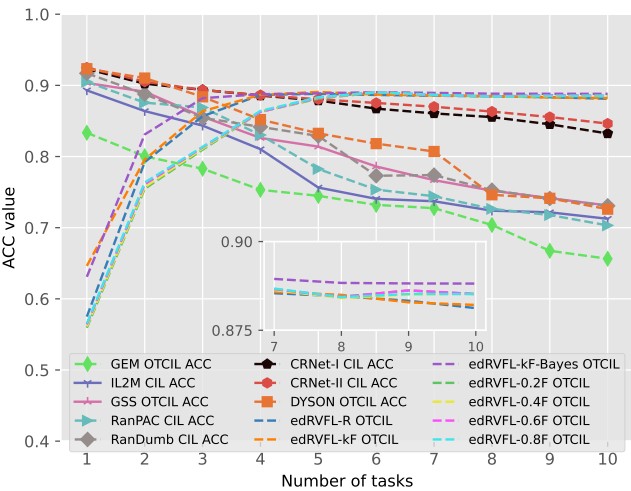

Figure 5: $ACC$ curves of methods on CIFAR-100/10 in CIL. Each task contained 10 classes. The edRVFL-R, edRVFL-$k$F, and edRVFL-$k$F-*Bayes* learned tasks in OTCIL. X-axis is locally enlarged in the inlaid subfigures.

## 5    CONCLUSION

In this study, we introduce *Variable Forward regularization* (-$k$F) as an enhancement to the existing *Forward regularization* (-F) for online linear learners. Our findings indicate that -F, while theoretically promising, underperforms in practice due to inappropriate intervention penalties and

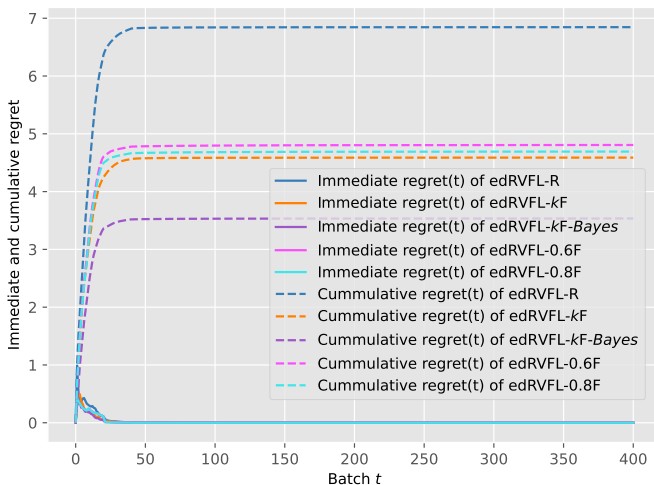

Figure 6: The CIFAR-100/10 testset immediate $regret(t)$ curves of edRVFL using -R, -$k$F, and -$k$F-*Bayes* strategies in OTCIL process. -$k$F-*Bayes* has the minimum loss in the whole process. X-axis denotes batch number, and Y-axis is regret and cumulative regret.

challenges posed by non-i.i.d nature in online learning. By modulating the intensity of -F with a variable $k$, -$k$F achieves improved stability and lower relative regret bounds, surpassing both -F and canonical *Ridge regularization* (-R). Additionally, we developed -$k$F-*Bayes* to dynamically adapt the penalty based on Bayesian principles, further addressing the instability issues. Experimental results in class incremental learning scenarios confirm the efficacy of our methods, highlighting their potential for reducing forgetting and enhancing performance in online learning tasks. Our future studies will focus on concrete regret bounds of -$k$F under stochastic setups.

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

## A APPENDIX

### A.1 THEOREM 2 PROOF

***Proof***: (7) can be expanded to:

$$\boldsymbol{\theta}_{t+1} = argmin_{\boldsymbol{\theta}} U_t(\boldsymbol{\theta}) - U_t(\boldsymbol{\theta}_t) + \mathcal{L}_t(\boldsymbol{\theta}) - (\boldsymbol{\theta} - \boldsymbol{\theta}_t)^T \nabla U_t(\boldsymbol{\theta}_t) + k \cdot \hat{\mathcal{L}}_{t+1}(\boldsymbol{\theta}) - k \cdot \hat{\mathcal{L}}_t(\boldsymbol{\theta}) \quad (23)$$

The $\nabla U_t(\boldsymbol{\theta}_t) = 0$ because of the latest convex optimization. (23) can be simplified to:

$$\boldsymbol{\theta}_{t+1} = argmin_{\boldsymbol{\theta}} U_{t+1}(\boldsymbol{\theta}) - U_t(\boldsymbol{\theta}_t) = argmin_{\boldsymbol{\theta}} \Delta_{U_0}(\boldsymbol{\theta}, \boldsymbol{\theta}_0) + \mathcal{L}_{1..t}(\boldsymbol{\theta}) + k \cdot \hat{\mathcal{L}}_{t+1}(\boldsymbol{\theta}) - c. \quad (24)$$

Solutions to both models (i.e. online learner (7) and offline expert (24)) remain the same at each time point $t$ if designating the last target as Bregman divergence function. This also suggests that $-k$F suffers no learning dissipation compared to the offline expert, because they have similar optimum at each step.

Proof finished. $\qquad \square$

### A.2 THEOREM 3 PROOF

***Proof***: Concretizing Theorem 1 with the **Setup**. For $0 \leq t \leq T$ we have:

$$\boldsymbol{\theta}_{t+1} = argmin_{\boldsymbol{\theta}} U_{t+1}(\boldsymbol{\theta}) \quad (25)$$

$$U_{t+1}(\theta) = \frac{1}{2}(\boldsymbol{\theta} - \boldsymbol{\theta}_0)^T \boldsymbol{\eta}_0^{-1}(\boldsymbol{\theta} - \boldsymbol{\theta}_0) + \sum_{i=1}^t \frac{1}{2}||\boldsymbol{x}_i^T \boldsymbol{\theta} - y_i||_2^2 + \frac{k}{2}||\boldsymbol{x}_{t+1}^T(\boldsymbol{\theta} - \boldsymbol{\theta}_0)||_2^2$$

Convert (25) into the form of (7) described by Theorem 2 to avoid the retrospective retraining, and derive variable learning rate via Lemma 2:

$$U_t(\boldsymbol{\theta}) + \frac{1}{2}||\boldsymbol{x}_t^T \boldsymbol{\theta} - y_t||_2^2 + \frac{k}{2}||\boldsymbol{x}_{t+1}^T(\boldsymbol{\theta} - \boldsymbol{\theta}_0)||_2^2 - \frac{k}{2}||\boldsymbol{x}_t^T(\boldsymbol{\theta} - \boldsymbol{\theta}_0)||_2^2 \quad (26)$$

$$= U_0(\boldsymbol{\theta}) - U_0(\boldsymbol{\theta}_0) - (\boldsymbol{\theta} - \boldsymbol{\theta}_0)^T \nabla U_0(\boldsymbol{\theta}_0) + \sum_{i=1}^t \frac{1}{2}||\boldsymbol{x}_i^T \boldsymbol{\theta} - y_i||_2^2 + \frac{k}{2}||\boldsymbol{x}_{t+1}^T(\boldsymbol{\theta} - \boldsymbol{\theta}_0)||_2^2$$

$$\Rightarrow U_t(\boldsymbol{\theta}) - U_0(\boldsymbol{\theta}) = \sum_{i=1}^{t-1} \frac{1}{2}||\boldsymbol{x}_i^T \boldsymbol{\theta} - y_i||_2^2 + \frac{k}{2}||\boldsymbol{x}_t^T(\boldsymbol{\theta} - \boldsymbol{\theta}_0)||_2^2 - \frac{1}{2}\boldsymbol{\theta}_0^T \boldsymbol{\eta}_0^{-1}\boldsymbol{\theta}_0 - (\boldsymbol{\theta} - \boldsymbol{\theta}_0)^T \boldsymbol{\eta}_0^{-1}\boldsymbol{\theta}_0$$

$$\Rightarrow U_t(\boldsymbol{\theta}) - U_0(\boldsymbol{\theta}) = \sum_{i=1}^{t-1} \frac{1}{2}||\boldsymbol{x}_i^T \boldsymbol{\theta} - y_i||_2^2 + \frac{k}{2}||\boldsymbol{x}_t^T(\boldsymbol{\theta} - \boldsymbol{\theta}_0)||_2^2 + \frac{1}{2}\boldsymbol{\theta}_0^T \boldsymbol{\eta}_0^{-1}\boldsymbol{\theta}_0 - \boldsymbol{\theta}^T \boldsymbol{\eta}_0^{-1}\boldsymbol{\theta}_0$$

$$\Rightarrow U_t(\boldsymbol{\theta}) - U_0(\boldsymbol{\theta}) = \frac{1}{2}\boldsymbol{\theta}^T \sum_{i=1}^{t-1} \boldsymbol{x}_i \boldsymbol{x}_i^T \boldsymbol{\theta} + \frac{1}{2}\sum_{i=1}^{t-1} y_i^2 - \sum_{i=1}^{t-1} y_i \boldsymbol{x}_i^T \boldsymbol{\theta}$$

$$- \boldsymbol{\theta}^T \boldsymbol{\eta}_0^{-1}\boldsymbol{\theta}_0 + \frac{k}{2}(\boldsymbol{\theta} - \boldsymbol{\theta}_0)^T \boldsymbol{x}_t \boldsymbol{x}_t^T(\boldsymbol{\theta} - \boldsymbol{\theta}_0) + \frac{1}{2}\boldsymbol{\theta}_0^T \boldsymbol{\eta}_0^{-1}\boldsymbol{\theta}_0$$

Based on Lemma 2, (26) can be simplified to:

$$U_t(\boldsymbol{\theta}) - (U_0(\boldsymbol{\theta}) + \frac{1}{2}\boldsymbol{\theta}^T \sum_{i=1}^{t-1} \boldsymbol{x}_i \boldsymbol{x}_i^T \boldsymbol{\theta} + \frac{k}{2}\boldsymbol{\theta}^T \boldsymbol{x}_t \boldsymbol{x}_t^T \boldsymbol{\theta}) = \boldsymbol{\omega}^T \boldsymbol{\theta} + c. \quad (27)$$

where $\boldsymbol{\omega} \in \mathbb{R}^d$. If set $V_t = \frac{1}{2}\boldsymbol{\theta}^T(\sum_{i=1}^{t-1} \boldsymbol{x}_i \boldsymbol{x}_i^T + k \cdot \boldsymbol{x}_t \boldsymbol{x}_t^T)\boldsymbol{\theta}$, based on (27) and Lemma 1 we have:

$$\Delta_{U_t}(\boldsymbol{\theta}, \boldsymbol{\theta}_t) = \Delta_{U_0 + V_t}(\boldsymbol{\theta}, \boldsymbol{\theta}_t) = \Delta_{U_0}(\boldsymbol{\theta}, \boldsymbol{\theta}_t) + \Delta_{V_t}(\boldsymbol{\theta}, \boldsymbol{\theta}_t) \quad (28)$$

$$= \frac{1}{2}(\boldsymbol{\theta} - \boldsymbol{\theta}_t)^T [\boldsymbol{\eta}_0^{-1} + \sum_{i=1}^{t-1} \boldsymbol{x}_i \boldsymbol{x}_i^T + k \cdot \boldsymbol{x}_t \boldsymbol{x}_t^T](\boldsymbol{\theta} - \boldsymbol{\theta}_t)$$

(28) shows that it alleviates the previous replay with all that knowledge absorbed in Bregman divergence. The stepwise solution to (28) in OL will change with $\boldsymbol{\eta}_t^{-1} = \boldsymbol{\eta}_0^{-1} + \sum_{i=1}^{t-1} \boldsymbol{x}_i \boldsymbol{x}_i^T + k \cdot \boldsymbol{x}_t \boldsymbol{x}_t^T$, also termed as the variable learning rate. The essence of OL algorithm using $-k$F is also to give time-varying optimization targets to guide network optimum (a posterior from the Bayesian view) updates on task streams. This optimization objective in Theorem 2 can be expressed further as:

$$\boldsymbol{\theta}_{t+1} = argmin_{\boldsymbol{\theta}} \Delta_{U_t}(\boldsymbol{\theta}, \boldsymbol{\theta}_t) + \mathcal{L}_t(\boldsymbol{\theta}) + k \cdot \hat{\mathcal{L}}_{t+1}(\boldsymbol{\theta}) - k \cdot \hat{\mathcal{L}}_t(\boldsymbol{\theta}) \quad (29)$$

$$= argmin_{\boldsymbol{\theta}} \frac{1}{2}(\boldsymbol{\theta} - \boldsymbol{\theta}_t)^T [\boldsymbol{\eta}_0^{-1} + \sum_{i=1}^{t-1} \boldsymbol{x}_i \boldsymbol{x}_i^T + k \cdot \boldsymbol{x}_t \boldsymbol{x}_t^T](\boldsymbol{\theta} - \boldsymbol{\theta}_t)$$

$$+ \frac{1}{2}||\boldsymbol{x}_t^T \boldsymbol{\theta} - y_t||_2^2 + \frac{k}{2}||\boldsymbol{x}_{t+1}^T(\boldsymbol{\theta} - \boldsymbol{\theta}_0)||_2^2 - \frac{k}{2}||\boldsymbol{x}_t^T(\boldsymbol{\theta} - \boldsymbol{\theta}_0)||_2^2$$

(29) can be transformed to the following constrained optimization problem:

$$\min_{\boldsymbol{\theta} \in \boldsymbol{\theta}_{t+1}} \frac{1}{2}(\boldsymbol{\theta} - \boldsymbol{\theta}_t)^T \boldsymbol{\eta}_t^{-1}(\boldsymbol{\theta} - \boldsymbol{\theta}_t) + \frac{1}{2}||\xi_1||_2^2 + \frac{k}{2}||\xi_2||_2^2 - \frac{k}{2}||\xi_3||_2^2 \tag{30}$$

$$s.t. \ \boldsymbol{x}_t^T \boldsymbol{\theta} - y_t = \xi_1; \boldsymbol{x}_{t+1}^T(\boldsymbol{\theta} - \boldsymbol{\theta}_0) = \xi_2; \boldsymbol{x}_t^T(\boldsymbol{\theta} - \boldsymbol{\theta}_0) = \xi_3, \forall t$$

The Lagrangian function of problem (30) is:

$$\ell(\boldsymbol{\theta}, \xi_{\{1,2,3\}}, \mu_{\{1,2,3\}}) = \frac{1}{2}(\boldsymbol{\theta} - \boldsymbol{\theta}_t)^T \boldsymbol{\eta}_t^{-1}(\boldsymbol{\theta} - \boldsymbol{\theta}_t) + \frac{1}{2}||\xi_1||_2^2 + \frac{k}{2}||\xi_2||_2^2 - \frac{k}{2}||\xi_3||_2^2 \tag{31}$$

$$+ \mu_1(\boldsymbol{x}_t^T \boldsymbol{\theta} - y_t - \xi_1) + \mu_2(\boldsymbol{x}_{t+1}^T(\boldsymbol{\theta} - \boldsymbol{\theta}_0) - \xi_2) + \mu_3(\boldsymbol{x}_t^T(\boldsymbol{\theta} - \boldsymbol{\theta}_0) - \xi_3)$$

The Lagrangian function (31) can be tackled through Karush-Kuhn-Tucker (KKT) conditions, which can be built into the following formulation:

$$\frac{\partial \ell(\boldsymbol{\theta}, \xi, \mu)}{\partial \boldsymbol{\theta}} = 0 \Rightarrow \boldsymbol{\eta}_t^{-1}(\boldsymbol{\theta} - \boldsymbol{\theta}_t) + \mu_1 \boldsymbol{x}_t + \mu_2 \boldsymbol{x}_{t+1} + \mu_3 \boldsymbol{x}_t = 0$$

$$\frac{\partial \ell(\boldsymbol{\theta}, \xi, \mu)}{\partial \xi_1} = 0 \Rightarrow \xi_1 = \mu_1$$

$$\frac{\partial \ell(\boldsymbol{\theta}, \xi, \mu)}{\partial \xi_2} = 0 \Rightarrow k\xi_2 = \mu_2$$

$$\frac{\partial \ell(\boldsymbol{\theta}, \xi, \mu)}{\partial \xi_3} = 0 \Rightarrow -k\xi_3 = \mu_3 \tag{32}$$

$$\frac{\partial \ell(\boldsymbol{\theta}, \xi, \mu)}{\partial \mu_1} = 0 \Rightarrow \boldsymbol{x}_t^T \boldsymbol{\theta} - y_t = \xi_1$$

$$\frac{\partial \ell(\boldsymbol{\theta}, \xi, \mu)}{\partial \mu_2} = 0 \Rightarrow \boldsymbol{x}_{t+1}^T(\boldsymbol{\theta} - \boldsymbol{\theta}_0) = \xi_2$$

$$\frac{\partial \ell(\boldsymbol{\theta}, \xi, \mu)}{\partial \mu_3} = 0 \Rightarrow \boldsymbol{x}_t^T(\boldsymbol{\theta} - \boldsymbol{\theta}_0) = \xi_3$$

Based on (32), we can obtain the recursive updating policy between $\boldsymbol{\theta}_t$ and $\boldsymbol{\theta}_{t+1}$ as follows:

$$\boldsymbol{\theta}_{t+1} = \boldsymbol{\theta}_t - \boldsymbol{\eta}_{t+1}[(\boldsymbol{x}_t \boldsymbol{x}_t^T + k \cdot \boldsymbol{x}_{t+1} \boldsymbol{x}_{t+1}^T - k \cdot \boldsymbol{x}_t \boldsymbol{x}_t^T)\boldsymbol{\theta}_t - \boldsymbol{x}_t y_t] \tag{33}$$

$$+ k \cdot \boldsymbol{\eta}_{t+1}(\boldsymbol{x}_{t+1} \boldsymbol{x}_{t+1}^T - \boldsymbol{x}_t \boldsymbol{x}_t^T)\boldsymbol{\theta}_0$$

Without loss of generality, given $\boldsymbol{\theta}_0 = 0$, to allow the learner to start from a blank model (no prior knowledge), (33) is rewritten as:

$$\boldsymbol{\theta}_{t+1} = \boldsymbol{\theta}_t - \boldsymbol{\eta}_{t+1}[(\boldsymbol{x}_t \boldsymbol{x}_t^T + k \cdot \boldsymbol{x}_{t+1} \boldsymbol{x}_{t+1}^T - k \cdot \boldsymbol{x}_t \boldsymbol{x}_t^T)\boldsymbol{\theta}_t - \boldsymbol{x}_t y_t] \tag{34}$$

The variable learning rate $\boldsymbol{\eta}^{-1}$ is also updated step-wise by using Sherman–Morrison–Woodbury law, as shown in (9).
Proof finished. $\qquad\qquad\square$

## A.3 REMARK 2 PROOF

***Proof***: (34) can be rewritten as:

$$\boldsymbol{\theta}_{t+1} = \boldsymbol{\theta}_t - \boldsymbol{\eta}_{t+1}[(\boldsymbol{\eta}_{t+1}^{-1} - \boldsymbol{\eta}_t^{-1})\boldsymbol{\theta}_t - \boldsymbol{x}_t y_t] \Rightarrow (12) \tag{35}$$

(12) can be rewritten as:

$$\boldsymbol{\eta}_t \boldsymbol{\eta}_{t+1}^{-1} \boldsymbol{\theta}_{t+1} = \boldsymbol{\theta}_t + \boldsymbol{\eta}_t \boldsymbol{x}_t y_t$$

$$\boldsymbol{\eta}_t(\boldsymbol{\eta}_t^{-1} + (1 - k) \cdot \boldsymbol{x}_t \boldsymbol{x}_t^T + k \cdot \boldsymbol{x}_{t+1} \boldsymbol{x}_{t+1}^T)\boldsymbol{\theta}_{t+1} = \boldsymbol{\theta}_t + \boldsymbol{\eta}_t \boldsymbol{x}_t y_t \Rightarrow (13) \tag{36}$$

Take the derivation of 25 to calculate $\boldsymbol{\theta}_{T+1}$ when $t = T$:

$$\frac{\partial U_{T+1}(\boldsymbol{\theta})}{\partial \boldsymbol{\theta}}\bigg|_{\boldsymbol{\theta} = \boldsymbol{\theta}_{T+1}} = 0 \Rightarrow \boldsymbol{\theta}_{T+1} = \boldsymbol{\eta}_{T+1} \sum_{i=1}^{T} \boldsymbol{x}_i y_i \Rightarrow (14) \tag{37}$$

Proof finished. $\qquad\qquad\square$

### A.4 THEOREM 4 PROOF

***Proof***: For $0 \le t \le T$, we expand the divergence $\Delta_{U_{t+1}}(\boldsymbol{\theta}, \boldsymbol{\theta}_{t+1})$:

$$\Delta_{U_{t+1}}(\boldsymbol{\theta}, \boldsymbol{\theta}_{t+1}) = U_{t+1}(\boldsymbol{\theta}) - U_{t+1}(\boldsymbol{\theta}_{t+1}) - (\boldsymbol{\theta} - \boldsymbol{\theta}_{t+1})^T \nabla_{\boldsymbol{\theta}} U_{t+1}(\boldsymbol{\theta}_{t+1}) \tag{38}$$
$$= U_{t+1}(\boldsymbol{\theta}) - U_{t+1}(\boldsymbol{\theta}_{t+1})$$

According to Theorem 2, we have:

$$U_{t+1}(\boldsymbol{\theta}) = U_t(\boldsymbol{\theta}) + \mathcal{L}_t(\boldsymbol{\theta}) + k \cdot \hat{\mathcal{L}}_{t+1}(\boldsymbol{\theta}) - k \cdot \hat{\mathcal{L}}_t(\boldsymbol{\theta}) \tag{39}$$

Substitute (38) into (39), for $1 \le t \le T$, we get:

$$\mathcal{L}_t(\boldsymbol{\theta}) = \Delta_{U_{t+1}}(\boldsymbol{\theta}, \boldsymbol{\theta}_{t+1}) + U_{t+1}(\boldsymbol{\theta}_{t+1}) - U_t(\boldsymbol{\theta}) - k \cdot \hat{\mathcal{L}}_{t+1}(\boldsymbol{\theta}) + k \cdot \hat{\mathcal{L}}_t(\boldsymbol{\theta}) \tag{40}$$

Let $\boldsymbol{\theta} = \boldsymbol{\theta}_t$ in (40), we obtain:

$$\mathcal{L}_t(\boldsymbol{\theta}_t) = \Delta_{U_{t+1}}(\boldsymbol{\theta}_t, \boldsymbol{\theta}_{t+1}) + U_{t+1}(\boldsymbol{\theta}_{t+1}) - U_t(\boldsymbol{\theta}_t) - k \cdot \hat{\mathcal{L}}_{t+1}(\boldsymbol{\theta}_t) + k \cdot \hat{\mathcal{L}}_t(\boldsymbol{\theta}_t) \tag{41}$$

Subtract (40) from (41), we have:

$$\mathcal{L}_t(\boldsymbol{\theta}_t) - \mathcal{L}_t(\boldsymbol{\theta}) = \Delta_{U_{t+1}}(\boldsymbol{\theta}_t, \boldsymbol{\theta}_{t+1}) - k \cdot \hat{\mathcal{L}}_{t+1}(\boldsymbol{\theta}_t) + k \cdot \hat{\mathcal{L}}_t(\boldsymbol{\theta}_t) \tag{42}$$
$$+ k \cdot \hat{\mathcal{L}}_{t+1}(\boldsymbol{\theta}) - k \cdot \hat{\mathcal{L}}_t(\boldsymbol{\theta}) - \Delta_{U_{t+1}}(\boldsymbol{\theta}, \boldsymbol{\theta}_{t+1}) - U_t(\boldsymbol{\theta}_t) + U_t(\boldsymbol{\theta})$$

Substitute (38) into (42) again, we obtain:

$$\mathcal{L}_t(\boldsymbol{\theta}_t) - \mathcal{L}_t(\boldsymbol{\theta}) = \Delta_{U_{t+1}}(\boldsymbol{\theta}_t, \boldsymbol{\theta}_{t+1}) - k \cdot \hat{\mathcal{L}}_{t+1}(\boldsymbol{\theta}_t) + k \cdot \hat{\mathcal{L}}_t(\boldsymbol{\theta}_t) \tag{43}$$
$$+ k \cdot \hat{\mathcal{L}}_{t+1}(\boldsymbol{\theta}) - k \cdot \hat{\mathcal{L}}_t(\boldsymbol{\theta}) - \Delta_{U_{t+1}}(\boldsymbol{\theta}, \boldsymbol{\theta}_{t+1}) + \Delta_{U_t}(\boldsymbol{\theta}, \boldsymbol{\theta}_t)$$

Integrate both sides of (43) over $1 \le t \le T$ and subtract $\Delta_{U_0}(\boldsymbol{\theta}, \boldsymbol{\theta}_0)$, we have:

$$\sum_{t=1}^{T} \mathcal{L}_t(\boldsymbol{\theta}_t) - (\Delta_{U_0}(\boldsymbol{\theta}, \boldsymbol{\theta}_0) + \sum_{t=1}^{T} \mathcal{L}_t(\boldsymbol{\theta})) = \sum_{t=1}^{T} [\Delta_{U_{t+1}}(\boldsymbol{\theta}_t, \boldsymbol{\theta}_{t+1}) - k \cdot \hat{\mathcal{L}}_{t+1}(\boldsymbol{\theta}_t) + k \cdot \hat{\mathcal{L}}_t(\boldsymbol{\theta}_t)]$$
$$+ k \cdot \hat{\mathcal{L}}_{T+1}(\boldsymbol{\theta}) - k \cdot \hat{\mathcal{L}}_1(\boldsymbol{\theta}) - \Delta_{U_{T+1}}(\boldsymbol{\theta}, \boldsymbol{\theta}_{T+1}) + \Delta_{U_1}(\boldsymbol{\theta}, \boldsymbol{\theta}_1) - \Delta_{U_0}(\boldsymbol{\theta}, \boldsymbol{\theta}_0) \tag{44}$$

Note that the last few terms in (44) are iteratively canceled out.

Let $t = 0$ in (39) and $\boldsymbol{\theta}_1 = \boldsymbol{\theta}_0 = 0$, we have $U_1(\boldsymbol{\theta}) = U_0(\boldsymbol{\theta}) + k \cdot \hat{\mathcal{L}}_1(\boldsymbol{\theta})$.

$$\Delta_{U_1}(\boldsymbol{\theta}, \boldsymbol{\theta}_1) - \Delta_{U_0}(\boldsymbol{\theta}, \boldsymbol{\theta}_0) = U_1(\boldsymbol{\theta}) - U_1(\boldsymbol{\theta}_1) - U_0(\boldsymbol{\theta}) + U_0(\boldsymbol{\theta}_0) = k \cdot \hat{\mathcal{L}}_1(\boldsymbol{\theta}) \tag{45}$$

Finally, we can obtain:

$$\sum_{t=1}^{T} \mathcal{L}_t(\boldsymbol{\theta}_t) - (\Delta_{U_0}(\boldsymbol{\theta}, \boldsymbol{\theta}_0) + \sum_{t=1}^{T} \mathcal{L}_t(\boldsymbol{\theta})) \tag{46}$$
$$= \sum_{t=1}^{T} [\Delta_{U_{t+1}}(\boldsymbol{\theta}_t, \boldsymbol{\theta}_{t+1}) - k \cdot \hat{\mathcal{L}}_{t+1}(\boldsymbol{\theta}_t) + k \cdot \hat{\mathcal{L}}_t(\boldsymbol{\theta}_t)] + k \cdot \hat{\mathcal{L}}_{T+1}(\boldsymbol{\theta}) - \Delta_{U_{T+1}}(\boldsymbol{\theta}, \boldsymbol{\theta}_{T+1})$$

Proof finished. $\square$

### A.5 THEOREM 5 PROOF

***Proof***: Use Lemma 1 and Setups to concrete the right part of (15) in Theorem 4:

$$\sum_{t=1}^{T} [\Delta_{U_{t+1}}(\boldsymbol{\theta}_t, \boldsymbol{\theta}_{t+1}) - k \cdot \hat{\mathcal{L}}_{t+1}(\boldsymbol{\theta}_t) + k \cdot \hat{\mathcal{L}}_t(\boldsymbol{\theta}_t)] + k \cdot \hat{\mathcal{L}}_{T+1}(\boldsymbol{\theta}) - \Delta_{U_{T+1}}(\boldsymbol{\theta}, \boldsymbol{\theta}_{T+1}) \tag{47}$$

$$= \sum_{t=1}^{T} [\frac{1}{2}(\boldsymbol{\theta}_t - \boldsymbol{\theta}_{t+1})^T \boldsymbol{\eta}_{t+1}^{-1}(\boldsymbol{\theta}_t - \boldsymbol{\theta}_{t+1}) - \frac{k}{2}||\boldsymbol{x}_{t+1}^T \boldsymbol{\theta}_t||_2^2 + \frac{k}{2}||\boldsymbol{x}_t^T \boldsymbol{\theta}_t||_2^2]$$

$$- \frac{1}{2}(\boldsymbol{\theta} - \boldsymbol{\theta}_{T+1})^T \boldsymbol{\eta}_{T+1}^{-1}(\boldsymbol{\theta} - \boldsymbol{\theta}_{T+1}) + \frac{k}{2}||\boldsymbol{x}_{T+1}^T \boldsymbol{\theta}||_2^2$$

Substitute (34) (12) (13) into the first term of (47):

$$\frac{1}{2}(\boldsymbol{\theta}_t - \boldsymbol{\theta}_{t+1})^T \boldsymbol{\eta}_{t+1}^{-1}(\boldsymbol{\theta}_t - \boldsymbol{\theta}_{t+1}) \tag{48}$$

$$= \frac{1}{2}(\boldsymbol{\theta}_t - \boldsymbol{\theta}_{t+1})^T(k \cdot \boldsymbol{x}_{t+1}\boldsymbol{x}_{t+1}^T\boldsymbol{\theta}_t + (1-k) \cdot \boldsymbol{x}_t\boldsymbol{x}_t^T\boldsymbol{\theta}_t - \boldsymbol{x}_t y_t)$$

$$= \frac{1}{2}(\boldsymbol{\theta}_t - \boldsymbol{\theta}_{t+1})^T(-k \cdot \boldsymbol{x}_{t+1}\boldsymbol{x}_{t+1}^T\boldsymbol{\theta}_{t+1} + (1-k) \cdot \boldsymbol{x}_t\boldsymbol{x}_t^T\boldsymbol{\theta}_t - \boldsymbol{x}_t y_t)$$

$$+ \frac{1}{2}(\boldsymbol{\theta}_t - \boldsymbol{\theta}_{t+1})^T(k \cdot \boldsymbol{x}_{t+1}\boldsymbol{x}_{t+1}^T)(\boldsymbol{\theta}_t + \boldsymbol{\theta}_{t+1})$$

$$= \frac{1}{2}[(1-k) \cdot \boldsymbol{\eta}_t\boldsymbol{x}_t\boldsymbol{x}_t^T\boldsymbol{\theta}_{t+1} + k \cdot \boldsymbol{\eta}_t\boldsymbol{x}_{t+1}\boldsymbol{x}_{t+1}^T\boldsymbol{\theta}_{t+1} - \boldsymbol{\eta}_t\boldsymbol{x}_t y_t]^T$$

$$\times [(1-k) \cdot \boldsymbol{x}_t\boldsymbol{x}_t^T\boldsymbol{\theta}_t - k \cdot \boldsymbol{x}_{t+1}\boldsymbol{x}_{t+1}^T\boldsymbol{\theta}_{t+1} - \boldsymbol{x}_t y_t] + \frac{1}{2}(\boldsymbol{\theta}_t - \boldsymbol{\theta}_{t+1})^T(k \cdot \boldsymbol{x}_{t+1}\boldsymbol{x}_{t+1}^T)(\boldsymbol{\theta}_t + \boldsymbol{\theta}_{t+1})$$

$$= \frac{1}{2}[(1-k)^2\boldsymbol{\theta}_{t+1}^T\boldsymbol{x}_t\boldsymbol{x}_t^T\boldsymbol{\eta}_t\boldsymbol{x}_t\boldsymbol{x}_t^T\boldsymbol{\theta}_t - k(1-k) \cdot \boldsymbol{\theta}_{t+1}^T\boldsymbol{x}_t\boldsymbol{x}_t^T\boldsymbol{\eta}_t\boldsymbol{x}_{t+1}\boldsymbol{x}_{t+1}^T\boldsymbol{\theta}_{t+1}$$

$$- (1-k)\boldsymbol{\theta}_{t+1}^T\boldsymbol{x}_t\boldsymbol{x}_t^T\boldsymbol{\eta}_t\boldsymbol{x}_t y_t + k(1-k) \cdot \boldsymbol{\theta}_{t+1}^T\boldsymbol{x}_{t+1}\boldsymbol{x}_{t+1}^T\boldsymbol{\eta}_t\boldsymbol{x}_t\boldsymbol{x}_t^T\boldsymbol{\theta}_t$$

$$- k^2 \cdot \boldsymbol{\theta}_{t+1}^T\boldsymbol{x}_{t+1}\boldsymbol{x}_{t+1}^T\boldsymbol{\eta}_t\boldsymbol{x}_{t+1}\boldsymbol{x}_{t+1}^T\boldsymbol{\theta}_{t+1} - k \cdot \boldsymbol{\theta}_{t+1}^T\boldsymbol{x}_{t+1}\boldsymbol{x}_{t+1}^T\boldsymbol{\eta}_t\boldsymbol{x}_t y_t$$

$$- (1-k) \cdot y_t\boldsymbol{x}_t^T\boldsymbol{\eta}_t\boldsymbol{x}_t\boldsymbol{x}_t^T\boldsymbol{\theta}_t + k \cdot y_t\boldsymbol{x}_t^T\boldsymbol{\eta}_t\boldsymbol{x}_{t+1}\boldsymbol{x}_{t+1}^T\boldsymbol{\theta}_{t+1} + y_t^2\boldsymbol{x}_t^T\boldsymbol{\eta}_t\boldsymbol{x}_t]$$

$$+ k \cdot \frac{1}{2}\boldsymbol{\theta}_t^T\boldsymbol{x}_{t+1}\boldsymbol{x}_{t+1}^T\boldsymbol{\theta}_t - k \cdot \frac{1}{2}\boldsymbol{\theta}_{t+1}^T\boldsymbol{x}_{t+1}\boldsymbol{x}_{t+1}^T\boldsymbol{\theta}_{t+1}$$

Substitute (14) into the last two terms of (47) and let $\boldsymbol{\eta}_* = (\boldsymbol{\eta}_0^{-1} + \sum_{i=1}^{T} \boldsymbol{x}_i\boldsymbol{x}_i^T)^{-1} \Rightarrow \boldsymbol{\theta}^{off} - \boldsymbol{\theta}_{T+1} = k \cdot \boldsymbol{\eta}_*\boldsymbol{x}_{T+1}\boldsymbol{x}_{T+1}^T\boldsymbol{\theta}_{T+1}$, we obtain:

$$- \frac{1}{2}(\boldsymbol{\theta} - \boldsymbol{\theta}_{T+1})^T \boldsymbol{\eta}_{T+1}^{-1}(\boldsymbol{\theta} - \boldsymbol{\theta}_{T+1}) + \frac{k}{2}||\boldsymbol{x}_{T+1}^T\boldsymbol{\theta}||_2^2 \tag{49}$$

$$= -\frac{1}{2}k^2 \cdot \boldsymbol{\theta}_{T+1}^T\boldsymbol{x}_{T+1}\boldsymbol{x}_{T+1}^T\boldsymbol{\eta}_*\boldsymbol{\eta}_{T+1}^{-1}\boldsymbol{\eta}_*\boldsymbol{x}_{T+1}\boldsymbol{x}_{T+1}^T\boldsymbol{\theta}_{T+1}$$

$$+ \frac{k}{2}\boldsymbol{\theta}_{T+1}^T(\boldsymbol{I} + k \cdot \boldsymbol{\eta}_*\boldsymbol{x}_{T+1}\boldsymbol{x}_{T+1}^T)^T\boldsymbol{x}_{T+1}\boldsymbol{x}_{T+1}^T(\boldsymbol{I} + k \cdot \boldsymbol{\eta}_*\boldsymbol{x}_{T+1}\boldsymbol{x}_{T+1}^T)\boldsymbol{\theta}_{T+1}$$

$$= -\frac{1}{2}k^2 \cdot \boldsymbol{\theta}_{T+1}^T(\boldsymbol{x}_{T+1}\boldsymbol{x}_{T+1}^T)^2\boldsymbol{\eta}_*(\boldsymbol{I} + k \cdot \boldsymbol{\eta}_*\boldsymbol{x}_{T+1}\boldsymbol{x}_{T+1}^T)\boldsymbol{\theta}_{T+1}$$

$$+ \frac{k}{2}\boldsymbol{\theta}_{T+1}^T\boldsymbol{x}_{T+1}\boldsymbol{x}_{T+1}^T(\boldsymbol{I} + k \cdot \boldsymbol{\eta}_*\boldsymbol{x}_{T+1}\boldsymbol{x}_{T+1}^T)^2\boldsymbol{\theta}_{T+1}$$

$$= \frac{k}{2}\boldsymbol{\theta}_{T+1}^T\boldsymbol{x}_{T+1}\boldsymbol{x}_{T+1}^T(\boldsymbol{I} + k \cdot \boldsymbol{\eta}_*\boldsymbol{x}_{T+1}\boldsymbol{x}_{T+1}^T)\boldsymbol{\theta}_{T+1}$$

Substitute (48) and (49) back into (47), we get:

$$\sum_{t=1}^{T}[\Delta_{U_{t+1}}(\boldsymbol{\theta}_t, \boldsymbol{\theta}_{t+1}) - k \cdot \hat{\mathcal{L}}_{t+1}(\boldsymbol{\theta}_t) + k \cdot \hat{\mathcal{L}}_t(\boldsymbol{\theta}_t)] + k \cdot \hat{\mathcal{L}}_{T+1}(\boldsymbol{\theta}) - \Delta_{U_{T+1}}(\boldsymbol{\theta}, \boldsymbol{\theta}_{T+1}) \qquad (50)$$

$$= \frac{1}{2}\sum_{t=1}^{T}[(1-k)^2\boldsymbol{\theta}_{t+1}^T\boldsymbol{x}_t\boldsymbol{x}_t^T\boldsymbol{\eta}_t\boldsymbol{x}_t\boldsymbol{x}_t^T\boldsymbol{\theta}_t - k(1-k)\cdot\boldsymbol{\theta}_{t+1}^T\boldsymbol{x}_t\boldsymbol{x}_t^T\boldsymbol{\eta}_t\boldsymbol{x}_{t+1}\boldsymbol{x}_{t+1}^T\boldsymbol{\theta}_{t+1}$$

$$- (1-k)\boldsymbol{\theta}_{t+1}^T\boldsymbol{x}_t\boldsymbol{x}_t^T\boldsymbol{\eta}_t\boldsymbol{x}_t y_t + k(1-k)\cdot\boldsymbol{\theta}_{t+1}^T\boldsymbol{x}_{t+1}\boldsymbol{x}_{t+1}^T\boldsymbol{\eta}_t\boldsymbol{x}_t\boldsymbol{x}_t^T\boldsymbol{\theta}_t$$

$$- k^2\cdot\boldsymbol{\theta}_{t+1}^T\boldsymbol{x}_{t+1}\boldsymbol{x}_{t+1}^T\boldsymbol{\eta}_t\boldsymbol{x}_{t+1}\boldsymbol{x}_{t+1}^T\boldsymbol{\theta}_{t+1} - (1-k)\cdot y_t\boldsymbol{x}_t^T\boldsymbol{\eta}_t\boldsymbol{x}_t\boldsymbol{x}_t^T\boldsymbol{\theta}_t + y_t^2\boldsymbol{x}_t^T\boldsymbol{\eta}_t\boldsymbol{x}_t$$

$$+ k\cdot\boldsymbol{\theta}_t^T\boldsymbol{x}_t\boldsymbol{x}_t^T\boldsymbol{\theta}_t - k\cdot\boldsymbol{\theta}_{t+1}^T\boldsymbol{x}_{t+1}\boldsymbol{x}_{t+1}^T\boldsymbol{\theta}_{t+1}] + \frac{k}{2}\boldsymbol{\theta}_{T+1}^T\boldsymbol{x}_{T+1}\boldsymbol{x}_{T+1}^T(\boldsymbol{I} + k\cdot\boldsymbol{\eta}_*\boldsymbol{x}_{T+1}\boldsymbol{x}_{T+1}^T)\boldsymbol{\theta}_{T+1}$$

$$= \frac{1}{2}\sum_{t=1}^{T}[(1-k)^2\boldsymbol{\theta}_{t+1}^T\boldsymbol{x}_t\boldsymbol{x}_t^T\boldsymbol{\eta}_t\boldsymbol{x}_t\boldsymbol{x}_t^T\boldsymbol{\theta}_t - k(1-k)\cdot\boldsymbol{\theta}_{t+1}^T\boldsymbol{x}_t\boldsymbol{x}_t^T\boldsymbol{\eta}_t\boldsymbol{x}_{t+1}\boldsymbol{x}_{t+1}^T\boldsymbol{\theta}_{t+1}$$

$$- (1-k)\boldsymbol{\theta}_{t+1}^T\boldsymbol{x}_t\boldsymbol{x}_t^T\boldsymbol{\eta}_t\boldsymbol{x}_t y_t + k(1-k)\cdot\boldsymbol{\theta}_{t+1}^T\boldsymbol{x}_{t+1}\boldsymbol{x}_{t+1}^T\boldsymbol{\eta}_t\boldsymbol{x}_t\boldsymbol{x}_t^T\boldsymbol{\theta}_t$$

$$- k^2\cdot\boldsymbol{\theta}_{t+1}^T\boldsymbol{x}_{t+1}\boldsymbol{x}_{t+1}^T\boldsymbol{\eta}_t\boldsymbol{x}_{t+1}\boldsymbol{x}_{t+1}^T\boldsymbol{\theta}_{t+1} - (1-k)\cdot y_t\boldsymbol{x}_t^T\boldsymbol{\eta}_t\boldsymbol{x}_t\boldsymbol{x}_t^T\boldsymbol{\theta}_t + y_t^2\boldsymbol{x}_t^T\boldsymbol{\eta}_t\boldsymbol{x}_t]$$

$$+ \frac{k^2}{2}\boldsymbol{\theta}_{T+1}^T\boldsymbol{x}_{T+1}\boldsymbol{x}_{T+1}^T\boldsymbol{\eta}_*\boldsymbol{x}_{T+1}\boldsymbol{x}_{T+1}^T\boldsymbol{\theta}_{T+1}$$

Try to simplify (50) further. When $t = T$ and using Sherman–Morrison–Woodbury law, we have:

$$\frac{k^2}{2}\cdot\boldsymbol{\theta}_{T+1}^T\boldsymbol{x}_{T+1}\boldsymbol{x}_{T+1}^T\boldsymbol{\eta}_*\boldsymbol{x}_{T+1}\boldsymbol{x}_{T+1}^T\boldsymbol{\theta}_{T+1} - \frac{k^2}{2}\cdot\boldsymbol{\theta}_{T+1}^T\boldsymbol{x}_{T+1}\boldsymbol{x}_{T+1}^T\boldsymbol{\eta}_T\boldsymbol{x}_{T+1}\boldsymbol{x}_{T+1}^T\boldsymbol{\theta}_{T+1} \qquad (51)$$

$$= \frac{k^2}{2}\cdot\boldsymbol{\theta}_{T+1}^T\boldsymbol{x}_{T+1}\boldsymbol{x}_{T+1}^T[\boldsymbol{\eta}_* - \boldsymbol{\eta}_T]\boldsymbol{x}_{T+1}\boldsymbol{x}_{T+1}^T\boldsymbol{\theta}_{T+1}$$

$$= \frac{k^2}{2[1 + (1-k)\boldsymbol{x}_T^T\boldsymbol{\eta}_T\boldsymbol{x}_T]}\cdot\boldsymbol{\theta}_{T+1}^T\boldsymbol{x}_{T+1}\boldsymbol{x}_{T+1}^T[-(1-k)\boldsymbol{\eta}_T\boldsymbol{x}_T\boldsymbol{x}_T^T\boldsymbol{\eta}_T]\boldsymbol{x}_{T+1}\boldsymbol{x}_{T+1}^T\boldsymbol{\theta}_{T+1}$$

According to the property of the symmetric definite matrix, given $0 < k < 1$ and $\boldsymbol{x} \neq \boldsymbol{0}$, (51) $< 0$. So the initial two terms in (51) can be eliminated in (50) as we target the upper restrictions. The (50) can be rewritten as:

$$\sum_{t=1}^{T}[\Delta_{U_{t+1}}(\boldsymbol{\theta}_t, \boldsymbol{\theta}_{t+1}) - k \cdot \hat{\mathcal{L}}_{t+1}(\boldsymbol{\theta}_t) + k \cdot \hat{\mathcal{L}}_t(\boldsymbol{\theta}_t)] + k \cdot \hat{\mathcal{L}}_{T+1}(\boldsymbol{\theta}) - \Delta_{U_{T+1}}(\boldsymbol{\theta}, \boldsymbol{\theta}_{T+1}) \qquad (52)$$

$$= \frac{1}{2}\sum_{t=1}^{T}[(1-k)^2\boldsymbol{\theta}_{t+1}^T\boldsymbol{x}_t\boldsymbol{x}_t^T\boldsymbol{\eta}_t\boldsymbol{x}_t\boldsymbol{x}_t^T\boldsymbol{\theta}_t - k(1-k)\cdot\boldsymbol{\theta}_{t+1}^T\boldsymbol{x}_t\boldsymbol{x}_t^T\boldsymbol{\eta}_t\boldsymbol{x}_{t+1}\boldsymbol{x}_{t+1}^T\boldsymbol{\theta}_{t+1}$$

$$- (1-k)\boldsymbol{\theta}_{t+1}^T\boldsymbol{x}_t\boldsymbol{x}_t^T\boldsymbol{\eta}_t\boldsymbol{x}_t y_t + k(1-k)\cdot\boldsymbol{\theta}_{t+1}^T\boldsymbol{x}_{t+1}\boldsymbol{x}_{t+1}^T\boldsymbol{\eta}_t\boldsymbol{x}_t\boldsymbol{x}_t^T\boldsymbol{\theta}_t$$

$$- (1-k)\cdot y_t\boldsymbol{x}_t^T\boldsymbol{\eta}_t\boldsymbol{x}_t\boldsymbol{x}_t^T\boldsymbol{\theta}_t + y_t^2\boldsymbol{x}_t^T\boldsymbol{\eta}_t\boldsymbol{x}_t] - \frac{k^2}{2}\cdot\sum_{t=1}^{T-1}\boldsymbol{\theta}_{t+1}^T\boldsymbol{x}_{t+1}\boldsymbol{x}_{t+1}^T\boldsymbol{\eta}_t\boldsymbol{x}_{t+1}\boldsymbol{x}_{t+1}^T\boldsymbol{\theta}_{t+1}$$

Given $\mathcal{X} \sim \mathcal{P}_{\mathcal{X}}$ and $\{\boldsymbol{x}_t\}_{t=1}^T$ respects the i.i.d premise,

$$\mathbb{E}[(1-k)^2\boldsymbol{\theta}_{t+1}^T\boldsymbol{x}_t\boldsymbol{x}_t^T\boldsymbol{\eta}_t\boldsymbol{x}_t\boldsymbol{x}_t^T\boldsymbol{\theta}_t + k(1-k)\cdot\boldsymbol{\theta}_{t+1}^T\boldsymbol{x}_{t+1}\boldsymbol{x}_{t+1}^T\boldsymbol{\eta}_t\boldsymbol{x}_t\boldsymbol{x}_t^T\boldsymbol{\theta}_t] \qquad (53)$$

$$\leq \mathbb{E}[(1-k)^2\boldsymbol{\theta}_{t+1}^T\boldsymbol{x}_t\boldsymbol{x}_t^T\boldsymbol{\eta}_t\boldsymbol{x}_t\boldsymbol{x}_t^T\boldsymbol{\theta}_t + k(1-k)\cdot\boldsymbol{\theta}_{t+1}^T\boldsymbol{x}_{t+1}\boldsymbol{x}_{t+1}^T\boldsymbol{\eta}_t\boldsymbol{x}_{t+1}\boldsymbol{x}_t^T\boldsymbol{\theta}_{t+1}]$$

$$= \mathbb{E}[(1-k)\cdot\boldsymbol{\theta}_{t+1}^T\boldsymbol{x}_t\boldsymbol{x}_t^T\boldsymbol{\eta}_t\boldsymbol{x}_t\boldsymbol{x}_t^T\boldsymbol{\theta}_t]$$

$$\mathbb{E}[(1-k)\cdot\boldsymbol{\theta}_{t+1}^T\boldsymbol{x}_t\boldsymbol{x}_t^T\boldsymbol{\eta}_t\boldsymbol{x}_t\boldsymbol{x}_t^T\boldsymbol{\theta}_t - (1-k)\cdot y_t\boldsymbol{x}_t^T\boldsymbol{\eta}_t\boldsymbol{x}_t\boldsymbol{x}_t^T\boldsymbol{\theta}_t] \qquad (54)$$

$$= \mathbb{E}[(1-k)\cdot\boldsymbol{\theta}_{t+1}^T\boldsymbol{x}_t\boldsymbol{x}_t^T\boldsymbol{\eta}_t\boldsymbol{x}_t\boldsymbol{x}_t^T\boldsymbol{\theta}_t - (1-k)\cdot\boldsymbol{\theta}_{t+1}^T\boldsymbol{x}_t\boldsymbol{x}_t^T\boldsymbol{\eta}_t\boldsymbol{x}_t\boldsymbol{x}_t^T\boldsymbol{\theta}_t]$$

$$= 0$$

Substitute (53) and (54) back into (52), we obtain:

$$\mathbb{E}[\sum_{t=1}^{T}[\Delta_{U_{t+1}}(\boldsymbol{\theta}_t, \boldsymbol{\theta}_{t+1}) - k \cdot \hat{\mathcal{L}}_{t+1}(\boldsymbol{\theta}_t) + k \cdot \hat{\mathcal{L}}_t(\boldsymbol{\theta}_t)] + k \cdot \hat{\mathcal{L}}_{T+1}(\boldsymbol{\theta}) - \Delta_{U_{T+1}}(\boldsymbol{\theta}, \boldsymbol{\theta}_{T+1})] \tag{55}$$

$$= \mathbb{E}[\frac{1}{2}\sum_{t=1}^{T}[-k(1-k) \cdot \boldsymbol{\theta}_{t+1}^T \boldsymbol{x}_t \boldsymbol{x}_t^T \boldsymbol{\eta}_t \boldsymbol{x}_{t+1} \boldsymbol{x}_{t+1}^T \boldsymbol{\theta}_{t+1} - (1-k)\boldsymbol{\theta}_{t+1}^T \boldsymbol{x}_t \boldsymbol{x}_t^T \boldsymbol{\eta}_t \boldsymbol{x}_t y_t + y_t^2 \boldsymbol{x}_t^T \boldsymbol{\eta}_t \boldsymbol{x}_t]$$

$$- \frac{k^2}{2} \cdot \sum_{t=1}^{T-1} \boldsymbol{\theta}_{t+1}^T \boldsymbol{x}_{t+1} \boldsymbol{x}_{t+1}^T \boldsymbol{\eta}_t \boldsymbol{x}_{t+1} \boldsymbol{x}_{t+1}^T \boldsymbol{\theta}_{t+1}]$$

$$= \mathbb{E}[\frac{1}{2}\sum_{t=1}^{T}[-(1-k)\boldsymbol{x}_t^T \boldsymbol{\eta}_t \boldsymbol{x}_t y_t^2 + y_t^2 \boldsymbol{x}_t^T \boldsymbol{\eta}_t \boldsymbol{x}_t] - \frac{k^2}{2} \cdot \sum_{t=1}^{T-1} \boldsymbol{\theta}_{t+1}^T \boldsymbol{x}_{t+1} \boldsymbol{x}_{t+1}^T \boldsymbol{\eta}_t \boldsymbol{x}_{t+1} \boldsymbol{x}_{t+1}^T \boldsymbol{\theta}_{t+1}]$$

$$\leq \mathbb{E}[\frac{k}{2}\sum_{t=1}^{T} y_t^2 \boldsymbol{x}_t^T \boldsymbol{\eta}_t \boldsymbol{x}_t]$$

To prevent the model from potential attacks, in the adversarial setting, we assume $\boldsymbol{x}_t^T \boldsymbol{\theta}_t$ lies in $[-Y_m, Y_m]$ for -R style as shown in Lemma 4. Obviously, our general $-kF$ style removes this and avoids clipping or $Y_m$ updates. To constrain (55) further, according to the Matrix Spectral Theorem, it can be proved that $e_{max}(\boldsymbol{A}) = \sup(\boldsymbol{p}^T \boldsymbol{A} \boldsymbol{p}| \|\boldsymbol{p}\|_2 = 1)$ is a convex function of $\boldsymbol{A}$, where $e_{max}$ serves as the maximum eigenvalue of real-valued symmetric matrix $\boldsymbol{A}$, (55) can be transformed to:

$$\mathbb{E}[\frac{k}{2}\sum_{t=1}^{T} y_t^2 \boldsymbol{x}_t^T \boldsymbol{\eta}_t \boldsymbol{x}_t] \tag{56}$$

$$= \mathbb{E}[\frac{k}{2}\sum_{t=1}^{T} y_t^2 \|\boldsymbol{x}_t\|_2^2 \frac{\boldsymbol{x}_t^T}{\|\boldsymbol{x}_t\|_2} \boldsymbol{\eta}_t \frac{\boldsymbol{x}_t}{\|\boldsymbol{x}_t\|_2}]$$

$$\leq \mathbb{E}[\frac{k}{2} Y_m^2 X_m^2 d \sum_{t=1}^{T} e_{max}(\boldsymbol{\eta}_t)]$$

$$= \mathbb{E}[\frac{k}{2} Y_m^2 X_m^2 d \sum_{t=1}^{T} \inf[\frac{1}{e_{min}(\boldsymbol{\eta}_t^{-1})}]]$$

where $X_m = \max_{1 \leq t \leq T}\{\|\boldsymbol{x}_t\|_\infty\}, Y_m = \max_{1 \leq t \leq T}\{|y_t|\}$, and $\sup(e_{min}(\boldsymbol{\eta}_t^{-1} - \boldsymbol{\eta}_0^{-1})) \leq X_m^2(t - 1 + k)$ because $\sum_i e_i(\boldsymbol{\eta}_t^{-1} - \boldsymbol{\eta}_0^{-1}) = trace(\boldsymbol{\eta}_t^{-1} - \boldsymbol{\eta}_0^{-1})$. Continue to simplify (56), we obtain:

$$\mathbb{E}[\frac{k}{2} Y_m^2 X_m^2 d \sum_{t=1}^{T} \inf[\frac{1}{e_{min}(\boldsymbol{\eta}_t^{-1})}]] \tag{57}$$

$$\leq \mathbb{E}[\frac{k}{2} Y_m^2 X_m^2 d \int_0^T \frac{1}{\lambda + X_m^2(t - 1 + k)} dt]$$

$$= \mathbb{E}[\frac{k}{2} Y_m^2 d \int_0^T \frac{1}{t + \frac{\lambda}{X_m^2} + k - 1}]$$

$$= \frac{k}{2} Y_m^2 d In(1 + \frac{T X_m^2}{\lambda + (k-1) \cdot X_m^2})$$

Proof finished. □

## A.6 THEOREM 6 PROOF

**Proof**: To enable $-kF$ to adaptively mediate the Forward regularization intensity for robust performance in OL/CL, the hard constant $k$ is replaced with soft variables. We will study the dynamic

updating process and introduce an algorithm for generating time-varying $k$ values. (6) in Theorem 1 can be written as:

$$\boldsymbol{\theta}_{t+1} = argmin_{\boldsymbol{\theta}} U_{t+1}(\boldsymbol{\theta}) \tag{58}$$

$$U_{t+1}(\boldsymbol{\theta}) = \Delta_{U_0}(\boldsymbol{\theta}, \boldsymbol{\theta}_0) + \mathcal{L}_{1..t}(\boldsymbol{\theta}) + k_{t+1} \cdot \hat{\mathcal{L}}_{t+1}(\boldsymbol{\theta})$$

Based on (58), we represent Theorem 2 as follows:

$$\boldsymbol{\theta}_{t+1} = argmin_{\boldsymbol{\theta}} \Delta_{U_t}(\boldsymbol{\theta}, \boldsymbol{\theta}_t) + \mathcal{L}_t(\boldsymbol{\theta}) + k_{t+1} \cdot \hat{\mathcal{L}}_{t+1}(\boldsymbol{\theta}) - k_t \cdot \hat{\mathcal{L}}_t(\boldsymbol{\theta}) \tag{59}$$

Similar to the proof of Theorem 3, we can obtain:

$$\boldsymbol{\theta}_{t+1} = argmin_{\theta} \frac{1}{2}(\boldsymbol{\theta} - \boldsymbol{\theta}_t)^T [\boldsymbol{\eta}_0^{-1} + \sum_{i=1}^{t-1} \boldsymbol{x}_i \boldsymbol{x}_i^T + k_t \cdot \boldsymbol{x}_t \boldsymbol{x}_t^T](\boldsymbol{\theta} - \boldsymbol{\theta}_t) \tag{60}$$

$$+ \frac{1}{2}||\boldsymbol{x}_t^T \boldsymbol{\theta} - y_t||_2^2 + \frac{k_{t+1}}{2} \cdot ||\boldsymbol{x}_{t+1}^T(\boldsymbol{\theta} - \boldsymbol{\theta}_0)||_2^2 - \frac{k_t}{2} \cdot ||\boldsymbol{x}_t^T(\boldsymbol{\theta} - \boldsymbol{\theta}_0)||_2^2$$

where the novel variable learning rate $\boldsymbol{\eta}_t^{-1} = \boldsymbol{\eta}_0^{-1} + \sum_{i=1}^{t-1} \boldsymbol{x}_i \boldsymbol{x}_i^T + k_t \cdot \boldsymbol{x}_t \boldsymbol{x}_t^T$. (34) is refreshed to:

$$\boldsymbol{\theta}_{t+1} = \boldsymbol{\theta}_t - \boldsymbol{\eta}_{t+1}[(\boldsymbol{x}_t \boldsymbol{x}_t^T + k_{t+1} \cdot \boldsymbol{x}_{t+1} \boldsymbol{x}_{t+1}^T - k_t \cdot \boldsymbol{x}_t \boldsymbol{x}_t^T)\boldsymbol{\theta}_t - \boldsymbol{x}_t y_t] \tag{61}$$

where the novel variable learning rate $\boldsymbol{\eta}_{t+1}^{-1} = \boldsymbol{\eta}_0^{-1} + \sum_{i=1}^{t} \boldsymbol{x}_i \boldsymbol{x}_i^T + k_{t+1} \cdot \boldsymbol{x}_{t+1} \boldsymbol{x}_{t+1}^T$.

From the perspective of Bayesian learning:

$$p(\boldsymbol{\theta}_{t+1}|y_1, ...y_t) \propto p(y_t|\boldsymbol{\theta}_{t+1}, y_1, ...y_{t-1}) \cdot p(\boldsymbol{\theta}_{t+1}|y_1, ...y_{t-1}), \tag{62}$$

the OL/CL can be regarded as a process of consecutively updating the posterior distribution of learnable weights based on current empirical risk and the prior distribution.

In (60), the prior Gaussian distribution of $\boldsymbol{\theta}$ can be:

$$p(\boldsymbol{\theta}_{t+1}) \sim \mathcal{N}(\boldsymbol{\theta}_t, \boldsymbol{\eta}_t) \tag{63}$$

Such that estimated distribution:

$$p(\boldsymbol{x}_{t+1}^T \boldsymbol{\theta}_{t+1}) \sim \mathcal{N}(\boldsymbol{x}_{t+1}^T \boldsymbol{\theta}_t, \boldsymbol{x}_{t+1}^T \boldsymbol{\eta}_t \boldsymbol{x}_{t+1}) \tag{64}$$

$$p(\boldsymbol{x}_t^T \boldsymbol{\theta}_{t+1}) \sim \mathcal{N}(\boldsymbol{x}_t^T \boldsymbol{\theta}_t, \boldsymbol{x}_t^T \boldsymbol{\eta}_t \boldsymbol{x}_t)$$

Given $\boldsymbol{x}_t$ and $\boldsymbol{x}_{t+1}$ are drawn from $\mathcal{P}_{\mathcal{X}}$, the $k_{t+1}$ and $k_t$ can be:

$$k_{t+1} = k_t = \boldsymbol{x}_t^T \boldsymbol{\eta}_t \boldsymbol{x}_t \tag{65}$$

To avoid the zero solution and allow manual adjustment, (65) is modified as follows:

$$k_{t+1} = k_t = \kappa \cdot (\boldsymbol{x}_t^T \boldsymbol{\eta}_t \boldsymbol{x}_t + \sigma) \tag{66}$$

where $\sigma$ is a small positive number (e.g. $10^{-5}$) and $\kappa$ is a positive constant (e.g. 1.0). Proof finished. □

## A.7 OTHER RESULTS OF NUMERICAL SIMULATION

Figures are shown in Figure 7.

## A.8 SUPPLEMENTARY MATERIAL TO CASE 2

**Evaluation metrics**: We introduced 6 metrics that characterized immediate testset accuracy, incremental task accuracy, knowledge retention ability, degree of knowledge loss, immediate regret, and Kullback-Leibler divergence (KL).

*Average task accuracy* ($ACC$) is defined in CL literature as the average accuracy of all previously learned tasks.

$$ACC = \frac{1}{|Q|} \sum_{q=1}^{Q} R_{Q,q} \tag{67}$$

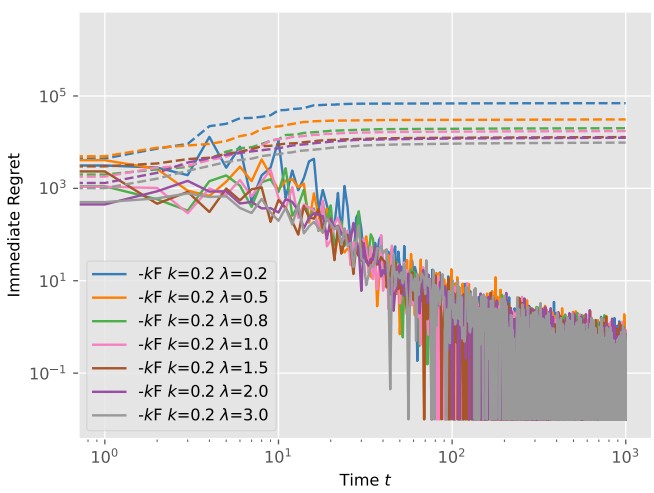

(a) Relative regrets of online learner using -0.2F.

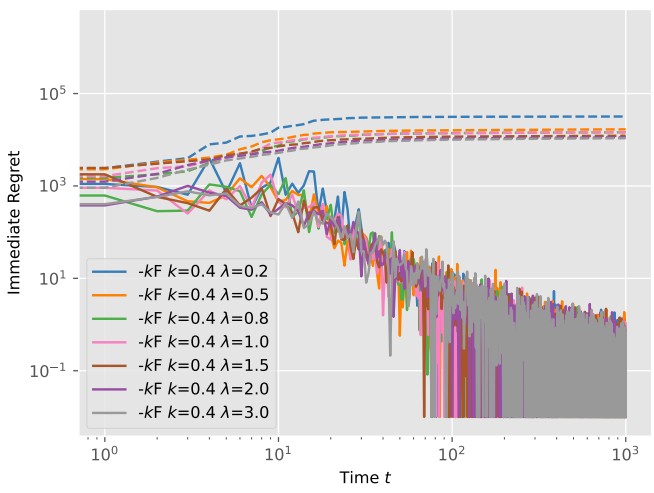

(b) Relative regrets of online learner using -0.4F.

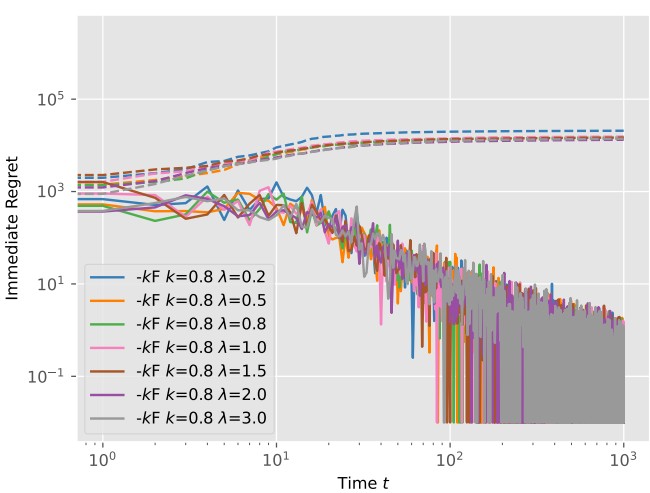

(c) Relative regrets of online learner using -0.8F.

19

Figure 7: Supplementary material to Figure 1.

Table 1: Description of 8 UCI tabular classification datasets

| DATASET | $|\mathcal{X}|$ | $|d|$ | $|\mathcal{Y}|$ |
|---|---|---|---|
| page blocks | 5473 | 10 | 5 |
| glass | 214 | 9 | 7 |
| image segmentation | 2310 | 9 | 7 |
| wine quality white | 4898 | 11 | 7 |
| pendigits | 10992 | 16 | 10 |
| yeast | 1484 | 8 | 10 |
| letters | 20000 | 16 | 26 |
| plant margin | 1600 | 64 | 100 |

where $R_{Q,q}$ is the classification accuracy of the learner on task $q$ after learning on task $Q(Q \geq q)$. It reflects the task-wise accuracy variation in both CIL and OTCIL processes.

*Backward transfer* ($BWT$) is defined as the average difference between the accuracy of all tasks completion and the first learning of one task:

$$BWT = \frac{1}{|Q| - 1} \sum_{q=1}^{Q-1} R_{Q,q} - R_{q,q} \tag{68}$$

BWT indicates the knowledge retention ability of the algorithm where larger values are desired.

*Forward transfer* ($FWT$) is defined as the average difference between the accuracy of the first learning of one task and using an independent expert on it:

$$FWT = \frac{1}{|Q| - 1} \sum_{q=2}^{Q} R_{q,q} - R_q^{ind} \tag{69}$$

where $R_q^{ind}$ denotes the testing accuracy of an independent expert trained only on task $q$. Higher FWT indicates the learner can acquire more knowledge from newly seen tasks.

*Immediate accuracy* ($acc.(t)$) denotes the immediate testing accuracy on entire testset after the learner finishing $t - th$ learning on $\mathcal{T}$ in OTCIL.

$$acc.(t) = R_{t,1..Q} \quad 1 \leq t \leq T \tag{70}$$

It is used to study the dynamic learning performance of using -$k$F and -$k$F-*Bayes* strategies precisely. Note that usually $acc.(T) \leq ACC$ for OTCIL algorithms.

*Immediate regret* ($regret(t)$) is defined as the real-time testset loss incurred by the learner:

$$regret(t) = ||\frac{\sum_{l=1}^{L} softmax(\boldsymbol{X}_{l,te}\boldsymbol{\theta}_{l,t}) - L \cdot \boldsymbol{Y}_{te}}{L \cdot |\mathcal{X}_{te}|}||_F^2 \tag{71}$$

where $1 \leq t \leq T$ and the subscript still denotes Frobenius norm. Also, we offer cumulative regret to describe the total incurred loss on testset in OTCIL process. Values may be given in logarithm.

Table 2: Testset accuracy comparison of CIL algorithms on 8 UCI tabular datasets

| SCENARIO | OUR OTCIL LEARNERS | | | | | | CIL METHODS | | | | OTCIL METHODS | |
| ALGORITHM | edRVFL-R†(w/o) | | edRVFL-kF†(w/o) | | edRVFL-kF-Bayes†(w/o) | | EWC(w/o) | | CRNet-I(w) | | DYSON†(w) | |
| DATASET | acc.(T)% | std.% | acc.(T)% | std.% | acc.(T)% | std.% | ACC% | std.% | ACC% | std.% | acc.(T)% | std.%(w) |
|---|---|---|---|---|---|---|---|---|---|---|---|---|
| page blocks | 95.68 | 0.34 | **96.08** | 0.15 | 95.90 | 0.08 | 90.13 | 0.78 | 91.15 | 0.96 | 91.52 | 0.59 |
| glass | 68.56 | 0.89 | 68.56 | 0.89 | **68.86** | 0.77 | 56.60 | 2.67 | 66.04 | 2.67 | 64.15 | 1.80 |
| image segmentation | 89.03 | 0.31 | 89.03 | 0.31 | 89.21 | 0.17 | 61.51 | 1.10 | 71.02 | 0.95 | 72.57 | 1.00 |
| wine quality white | 60.29 | 0.63 | 60.75 | 0.71 | 61.03 | 0.31 | 43.61 | 0.84 | 47.22 | 0.65 | 56.60 | 0.92 |
| pendigits | 96.33 | 0.08 | 96.89 | 0.08 | 97.51 | 0.14 | 79.40 | 0.60 | 81.36 | 0.73 | 78.74 | 1.24 |
| yeast | 59.79 | 0.46 | **59.97** | 0.46 | 58.81 | 0.23 | 45.42 | 1.05 | 49.73 | 0.89 | 52.34 | 0.90 |
| letters | 93.27 | 0.06 | 93.61 | 0.13 | **94.36** | 0.09 | 59.81 | 0.96 | 60.26 | 0.87 | 82.67 | 0.93 |
| plant margin | 77.92 | 0.29 | **80.38** | 0.81 | 79.83 | 0.93 | 38.75 | 1.68 | 36.50 | 1.93 | 53.77 | 2.03 |

Note: The † denotes this algorithm can be applied to OTCIL scenario; (w) or (w/o) indicates algorithms with or without using pre-trained models; higher $acc.(T)\%$ with lower $std.\%$ is better; the best accuracy of each dataset is shown in **bold**; the underline indicates that the edRVFL-kF-Bayes outperforms both edRVFL-R and edRVFL-kF.

