# OpenReview forum: "Variable Forward Regularization to Replace Ridge in Online Linear Regression"
_ICLR.cc/2025/Conference — ICLR 2025 Conference Withdrawn Submission_

### Official Review · Reviewer_BTEX · 2024-10-31

**Soundness:** 3
**Presentation:** 1
**Contribution:** 2
**Rating:** 3
**Confidence:** 5

**Summary:**

The authors observe that forward regularization (-F) cannot perform as expected in practice, even possibly losing to ridge regularization (-R) for online learning tasks. Based on this observation, this work identifies two main reasons and introduces Variable Forward regularization (-$k$F), which incorporates a variable $k$ into -F. The authors then apply their algorithm to online learning tasks and provide theoretical analysis. Finally, they also present an adaptive -$k$F, termed -$k$F-Bayes, and apply both -$k$F and -$k$F-Bayes to a class-incremental scenario in the experiments.

**Strengths:**

This work provides a significant finding in online linear regression problem and proposes a solution. The authors conduct comprehensive experiments to validate the effectiveness of their approach, contributing to the advancement of class-incremental continual learning.

**Weaknesses:**

The novelty of this work is limited, as their approach merely introduces a controllable variable $k$ into the previous method -F. Additionally, the writing in this paper needs improvement. The main weakness is the attempt to include an excessive number of lemmas and theorems without clearly explaining the relationships among these theoretical elements. Specifically, in Sections 3.1 to 3.3, the authors present five theorems, most of which lack a description of the algorithm and a discussion of the results. More confusingly, there is considerable inconsistency in the use of mathematical notation. For instance, none of the theorems or lemmas are placed within the `{theorem}` or `{lemma}` environments of LaTeX, and it should be $\arg\min$, not $argmin$. Most perplexingly, in Equation (4), what does $In$ mean? It should be $\ln$. Did the authors explain what $\theta^r_t$ and $d$ in (4) represents? Finally, the theoretical guarantee provided by this work does not improve the bound order.

I strongly encourage the authors to thoroughly revise their paper, as the current version is not reader-friendly.

**Questions:**

The authors claim that the proposed method achieves a tighter bound than -F in adversarial settings. However, from Theorem 5, it appears that -$k$F only achieves a regret bound of $O(d\log T)$, which is the same as the previous method. Therefore, my question is, how is the tighter bound obtained? Additionally, Remark 4 does not provide an explanation on how to select $k$; I would like to know, from a theoretical perspective, how $k$ should be chosen.

---

### Official Review · Reviewer_gQ4N · 2024-11-03

**Soundness:** 2
**Presentation:** 1
**Contribution:** 2
**Rating:** 3
**Confidence:** 4

**Summary:**

This paper considers a variant of the classic VAW forecaster in which the current feature’s contribution to the regularizer is multiplied by some value $k$. The standard value for $k$ is either 0 (which is the basic FTRL update), or $k=1$, which is the classic “improper” update employed by VAW.

The idea is that this change will enable better performance since the previous restricted values of $k$ may not be optimal.

**Strengths:**

The idea is interesting and potentially valuable, I have not seen this proposed before.

**Weaknesses:**

I have a number of concerns about presentation that I think are significant enough that I'd be cautious about accepting the paper based on them alone, although I am generally willing to be convinced that a paper's technical contribution is strong enough that people will overlook presentation issue. However, there are also some more serious technical concerns I detail later.

# presentation issues

This paper would benefit from some substantial revision regarding the presentation. Almost every paragraph contains grammatical errors, and also the authors seem to be confused about the difference between a Lemma/Theorem, a Definition, and an Algorithm. In brief, lemmas and theorems require proofs (unless the statement is so standard that the proof can be omitted), and definitions and algorithms do not. For example, Lemmas 1,3 are actually Definitions, while Lemmas 2/4 are indeed Lemmas. Some of these confusions result in technical language that is at best misleading and arguably actually incorrect: for example, on line 213 it is claimed that $\Gamma$ is a complex random variable. But $\Gamma$ is real-valued. I expect the authors instead wished to say that the distribution for $\Gamma$ is hard to understand.

The beginning of the paper claims that the paper will consider some “continual learning” setting. This setting is never properly defined, and moreover as far as I could tell it was never mentioned again in the rest of the paper. Instead, the results appear to be in the standard online linear regression setup exactly as considered in the classic VAW forecaster the authors are attempting to improve. I don’t understand what the point of the discussion about continual learning and catastrophic forgetting or non-stationarity was - it seems to be irrelevant.


I do not know what a “relative regret bound” is. It seems to be the same thing as what most literature just calls a “regret bound” without the word “relative”.


A few other specific (although minor) presentation issues that should be easy to fix: I think the authors wrote “In” using a capital i and a lowercase n to indicate the natural logarithm. The authors may wish to know that the correct way to indicate the natural logarithm is a lowercase L followed by a lowercase n: “ln”. Unfortunately, these look remarkably similar in many fonts, but they actually look very different in the font that was used in this paper. You can also use the latex built-in macro $\ln$.
I’m also not sure if the authors are aware of the standard Lemma/Theorem environments in Latex - it looks like these were written with just using "\textbf". There are plenty of tutorials on how to do this online.


These presentation issues are rather distracting, and likely would limit the impact of the paper if it were published. However, I do think it was still possible to follow the paper for the purposes of evaluating the technical content, although there are exceptions. For example, I frankly have very little idea what the paragraph from lines 240 to 244 is saying. I guessed that it was saying “we want to tune k online, but this might be hard”, but most of the words did not make sense to me.

# technical issues

One key problem with this paper is that the main Theorem (Theorem 5), seems to be obviously incorrect when $k$ is close to zero. Indeed, the limit as $k\to 0$ is at best 0 if $X_m^2<\lambda$, but possibly negative otherwise. This is clearly impossible. At first I thought the authors had mistakenly forgotten to stipulate that $k\ge 1$ or some similar restriction. However, I see that the authors actually only wanted to consider the scenario $k\in (0,1)$, which certainly contains problematic near-zero values for $k$.
That said, it is actually true that even with $k=0$ the regret should still be logarithmic in $T$ so long as $\lambda$ is sufficiently large. Perhaps there should be $k+1$ as the leading coefficient rather than $k$? Clearly some fix is required.


I also had a hard time understanding what was going on in equations 17-19. This seems to be an attempt to show that $k\ne 1$ can yield better bounds. This is definitely an important point to make, but I don’t follow it here. The authors do not explain what is meant by writing equation 19: obviously it does not *need* to hold that $k\in(0,1)$, so I guessed that the authors meant to write something like “from equations 17 and 18, we see that there is some $k\in(0,1)$ such that the regret bound for this $k$ is smaller than the regret bound with $k=1$”.  On a second reading, I could see that this is what the authors are attempting to argue by $(17)-(18)>0$, but this needs a much better explanation. Also, the stipulation that noise or feature vectors are gaussian seemed completely irrelevant to these calculations. Of course, this is all contingent on fixing Theorem 5.

I do not understand the Bayesian algorithm. The update for $k$ is completely unmotivated. Please explain what is happening here, why it is happening, and what if any theoretical result I can expect from this choice for $k$.

The caption of the plots claims that the plots are plotting “relative regret”, but the y-axis states “immediate regret”. What is “immediate regret”, and which is being plotted, relative or immediate regret?

If the point of the paper is to claim that $k$ strictly between zero and 1 is the best choice, why do none of the plots actually compare these choices? This seems like the only important kind of plot. I don’t see why we should care about varying $\lambda$ since this is possible no matter what $k$ is and so is not a novel part of the contribution. However, every plot seems to only vary $\lambda$.

**Questions:**

Can the technical issues be addressed?

---

### Official Review · Reviewer_ZuB2 · 2024-11-04

**Soundness:** 1
**Presentation:** 1
**Contribution:** 1
**Rating:** 1
**Confidence:** 4

**Summary:**

This paper studies some weighted modification of the Vovk-Azoury-Warmuth forecaster which takes an additional hyperparameter as the input. Then, a Bayesian heuristic strategy is proposed to select this hyperparameter, and the proposed algorithms are tested in experiments.

**Strengths:**

I'm sorry that I have to be harsh: this paper doesn't have any strength.

**Weaknesses:**

First, the quality of writing is poor. Throughout the paper the writing lacks the basic precision of scientific communication, which makes it very hard to extract the concrete arguments and verify their correctness. The proposed algorithm and the baselines (-F and -R) are never rigorously defined. For example, in line 150, what does it mean to have previous L concentrated into Bregman divergence? Such an imprecise statement can't appear in a theorem. There are many more issues like this.

Next, the theoretical result is flawed. Remark 3 does not make sense, and Remark 4 is incorrect. Somewhere in this remark it is assumed that the proposed algorithm is better than the baseline, but at the end this also appears as the conclusion, which is absurd. The theoretical advantage claimed by this paper is not supported by enough evidence.

Overall I see this paper as a clear rejection.

Minor: please also consider writing the natural log as \ln rather than In.

**Questions:**

N/A

---

### Official Review · Reviewer_qper · 2024-11-04

**Soundness:** 3
**Presentation:** 1
**Contribution:** 2
**Rating:** 1
**Confidence:** 3

**Summary:**

The paper proposes Variable Forward regularization (-kF), which interpolates between the ridge regularization (-R) and forward regularization (-F) by scaling the "hallucinating" term by some $k \geq 0$. The authors show an in-expectation regret upper bound for -kF, and proposes -kF-Bayes, which adaptively chooses $k$. The efficacy of the proposed algorithms are numerically verified in various continual learning (CL) scenarios.

**Strengths:**

- A simple idea with improved regret bounds compared to -F
- Experimentally, -kF and -kF-Bayes perform well for continual learning. (Disclaimer: I do not have the necessary knowledge on CL to determine whether the experiments are really good or not)

**Weaknesses:**

- IMHO, the writing of this paper needs significant improvement, as the current form hinders clear understanding. This was one of the reasons for my low score. Here are some points that came across my eyes (there are more, I believe):
   - A preliminary section on -F and -R would be nice to keep the paper self-contained. At least the authors should include the precise expressions for -F and -R in the Introduction when they are mentioned.
   - The **Lemma** and **Theorem** environments are overused to the point that even simple definitions are part of those environments. Thus it is very hard to tell exactly what the authors are trying to say in each theoretical result. Here are the problematic parts:
        - **Lemma 1**: definition of Bregman divergence
        - **Lemma 3**: Is this introducing some notations?
        - **Theorem 1**: What is proven here?
        - **Theorem 2**: Are you proving that (6) can be rewritten as (7)?
        - **Remark 2**: should be a lemma, if it is used in later proofs of the regrets??
        - **Theorem 6**: Exactly what have you proven here? In Appendix A.6, it seems that the choice of $k_t$ is derived under certain Bayesian setting.
   - Several undefined notations and typos: $\mathcal{L}_{1 \cdots Q}$, $In \rightarrow \ln$($\log$), in Eqn. (21), what do you mean by $k\_\{t+1\} = k_t$?, in Eqn. (22) what is $\kappa$?, in Eq. (30), $\theta \in \theta\_\{t+1\}$?,
   - Appropriate citations and credits should be given when citing known results. For instance, "Lemma 4" should cite Theorem 4.6 and 5.6 of Azoury & Warmuth (2001)
   - [my personal opinion] Given how the paper's considered setting does *not* go beyond the stochastic linear regression, I don't see any reason for overcomplicating the exposition by using Bregman divergence. The authors should consider rewriting everything in terms of Mahalanobis distances. This would make the theorem statements, proofs, and algorithms much more readable.
   - [my personal opinion] It would be nicer to denote the matrices using capital variables instead of lower cases, e.g., ${\bf \eta}_0 \rightarrow {\bf V}_0$.
   - It would be helpful to at least plot $k$ vs. Eqn. (16) to show the effect of $k$, and compare it with the actual regrets obtained by running the -kF.

- Overall, the organization of the paper is not yet ready for publication anywhere. The paper lacks qualitative discussions on all the theoretical results, consisting of list of lemmas, remarks and theorems.

- Several related works on forward regularization are missing:
   - [1] considers the *same* stochastic linear model and analyzes the high-probability regret of -R and -F
   - [2] considers nonstationary online learning (so quite different), yet the proposed discounted Vovk-Azoury-Warmuth forecaster resembles the weighting strategy proposed by the authors

- The plots are also very unclear regarding what they are trying to explain.


Overall, apart from the problems that I have with the theoretical results, I am giving the paper a score of 1 for its lack of readability, poor writing/organization, and lack of self-containedness. I will reconsider my score if the authors can significantly revise the paper during the rebuttal period to address all of my (and probably other reviewers') concerns.



[1] https://openreview.net/forum?id=rDdb26AQ0SO

[2] https://proceedings.mlr.press/v235/jacobsen24a.html

**Questions:**

- In the very beginning, the authors state that "we observe that -F could not perform as expected in experiments, even possibly failing to -R during OL". Is this solely based on the authors' presented experiments on CL?
    - If it is, I would say that this statement is quite misleading, as several results have shown that in many scenarios, replacing -R with -F is better; see, e.g., [1]. This question relates to Weakness 1.

- In the regret bounds of the paper, why do the authors consider infty-norm bounds on the decision space? To my knowledge, prior OL regret bounds rely on l2-norm bounds, which are tighter than the infty-norm bounds.

- Why is the definition of regret different for -R and -F? If I recall correctly, for both -R and -F, the regret definitions should be as in Eqn. (5) of the paper. What is Eqn. (4)?

- In Theorem 5, why do we require that $k > 0$? It seems that when $k = 0$, the regret is zero? This seems a bit weird, considering how the authors have claimed that -kF recovers -R when $k = 0$?

- At line 261-262, can the authors elaborate on why -kF-Bayes can resist catastrophic forgetting?

- In Appendix A.2, why do the authors at the end transform an unconstrained problem to a constrained one? Can't one just consider gradient descent on the unconstrained objective?

- The paper lacks a regret bound for -kF-Bayes, which seems crucial in providing the complete theoretical picture. Is this possible?

- [quite minor] Any connection to https://jmlr.org/papers/v24/22-0291.html ?

---

### Note · Authors · 2024-11-25

I have read and agree with the venue's withdrawal policy on behalf of myself and my co-authors.